# Characteristics of fine particle matters at the top of Shanghai Tower

Changqin Yin[1], Jianming Xu[1], Wei Gao[1], Liang Pan[1], Yixuan Gu[1], Qingyan Fu[2], Fan Yang[3]

[1]Shanghai Key Laboratory of Meteorology and Health, Shanghai, 200030, China
[2]Shanghai Environmental Monitoring Center, Shanghai, 200433, China
[3]Pudong New District Environmental Monitoring Station, Shanghai, 200032, China

*Correspondence to*: Jianming Xu (metxujm@163.com)

**Abstract.** To investigate the physical and chemical processes of fine particle matters at mid-upper planetary boundary layer (PBL), we conducted one-year continuous measurements of fine particle matters (PM), chemical composition of non-refractory submicron aerosol (NR-$PM_1$) and some gas species (including sulfur dioxide, nitrogen oxides and ozone) at an opening observatory (~600 m) at the top of Shanghai Tower (SHT), which is the Chinese 1[st] and World's 2[nd] highest building located in the typical financial central business district of Shanghai, China. This is the first report for the characteristics of fine particles based on continuous and sophisticated online measurements at the mid-upper level of urban PBL. The observed $PM_{2.5}$ and $PM_1$ mass concentrations at SHT were 25.5±17.7 and 17.3±11.7 μg m$^{-3}$ respectively. Organics, nitrate ($NO_3$) and sulfate ($SO_4$) occupied the first three leading contributions to NR-$PM_1$ at SHT, accounting for 35.8 %, 28.6 % and 20.8 % respectively. The lower $PM_{2.5}$ concentration was observed at SHT by 16.4 % compared with that near surface during the observation period. It was attributed to the decreased nighttime $PM_{2.5}$ concentrations (29.4 % lower than surface) at SHT in all seasons due to the complete isolations from both emissions and gas precursors near surface. However, daytime $PM_{2.5}$ concentrations at SHT were 12.4-35.1 % higher than those near surface from June to October, resulted from unexpected larger $PM_{2.5}$ levels during early to middle afternoon at SHT than surface. We suppose the significant chemical production of secondary aerosols existed in mid-upper PBL because strong solar irradiance, adequate gas precursors (e.g., NOx) and lower temperature were observed at SHT favorable for both photochemical production and gas-to-particle partitioning. This was further demonstrated by the significant increasing rate of oxygenated organic aerosols and $NO_3$ observed at SHT during 8:00-12:00 in spring (7.4 % h$^{-1}$ and 12.9 % h$^{-1}$), autumn (9.3 % h$^{-1}$ and 9.1 % h$^{-1}$) and summer (13.0 % h$^{-1}$ and 11.4 % h$^{-1}$), which cannot be fully explained by vertical mixing. It was noting that extremely high $NO_3$ was observed at SHT both in daytime and nighttime in winter, accounting for 37.2 % in NR-$PM_1$, suggesting the efficient pathway from heterogeneous and gas oxidated formation. Therefore, we highlight the priority of NOx reduction in Shanghai for the further improvement of air quality. This study reported greater daytime $PM_{2.5}$ concentrations at the height of 600 m in urban PBL compared with surface measurement, providing insight into their potential effects on local air quality, radiation forcing, and cloud/fog formations. We propose that the efficient production of secondary aerosol in mid-upper PBL should be cognized and explored more comprehensively by synergetic observations in future.

# 1 Introduction

The fine particle matters (PM) can absorb and scatter solar radiation, and act as cloud condensation nuclei. Thus, they can impact on Earth's energy budget directly and indirectly (Yu et al., 2006). Therefore, PM observations are important and necessary. Although worldwide surface (SUR) PM observation networks are reinforcing (Zhang and Cao, 2015; Solomon et al., 2014), the characterization of PM vertical distribution remains uncertain. The PM profiles can be acquired through ground-based lidar (Pappalardo et al., 2014), or airborne measurements (Kulmala et al., 2004). For aircraft observation, the advantage is a platform where flexible instruments can be equipped. For example, the aircraft studies gained aerosol size and composition during ACE-Asia field campaign with an Aerodyne aerosol mass spectrometer (AMS) (Bahreini et al., 2003). For lidar observation, the continuous long-term and high spatial resolution data can be achieved at the same time (Liu et al., 2021; Voudouri et al., 2020). Nevertheless, PM concentration retrieval based on lidar extinction coefficient highly depends on aerosol size distribution, aerosol composition, and atmospheric relative humidity assumptions, which are highly uncertain (Tao et al., 2016).

AMS technique is frequently applied in field observation to analyze PM chemical composition (including organics, nitrate, sulfate, ammonium, and chloride) (Frohlich et al., 2015; Zhang et al., 2007). Based on organic aerosol (OA) data of AMS, the source apportionment of OAs can be performed through positive matrix factorization (PMF) (Zhang et al., 2011). The common outcomes of OA PMF source apportionment are hydrocarbon-like and oxygenated OA (HOA and OOA, respectively). As part of ACE-Asia field campaign, the AMS was deployed in Asia for the first time (Zhou et al., 2020). As mentioned before, vertical PM composition observation needs a platform, which aircraft, mountain, tower, or high-altitude building can serve as. The observations conducted at mountain, tower and high-altitude building make up the "blind zone" of aircraft and lidar observations.

For aircraft observation, the height can reach free troposphere. Previous aircraft studies discovered OA formation in nighttime planetary boundary layer (PBL) (Brown et al., 2013; Pratt et al., 2012) and shallow cumulus clouds (Wonaschuetz et al., 2012). Besides, the distinct vertical distributions of PM chemical species were revealed (Brooks et al., 2019; Aldhaif et al., 2018; Liu et al., 2019). Zhao et al. (2020b) studied the vertical dispersion of size-resolved carbonaceous aerosols by comparing data at near surface level and hilltop. Based on volatile organic compound (VOC) measurements at the heights of 118m and 488m at Canton Tower, Mo et al. (2020) estimated the emission flux of VOC and secondary OA (SOA) formation potential using a mixed layer gradient technique. The measurements on a 300m research tower in a suburban area near Denver showed that the sampling site was under the influence of aged air masses at heights between 40m and 120m, while the fresh emissions below 40m (Ozturk et al., 2013). Based on a 325m meteorological tower in Beijing, previous researchers studied the vertical distribution of chemical species through a series of field campaigns (Chen et al., 2015; Zhao et al., 2020c). Zhou et al. (2018b) found that the differences between PM chemical species at SUR and 260m originated from the

different impacts of regional transport and local emission on primary and secondary species. Xie et al. (2019) presented the contribution of brown carbon to aerosol absorption.

Although previous studies made significant contribution to understanding PM vertical characteristics, long-term observations of PM and their chemical composition in the middle and upper boundary in high density residential area were in lack. Shanghai is one of the most densely populated megacities in the world. In this study, we present one-year continuous observation of $PM_{2.5}$ and $PM_1$ mass concentrations at the top of 632 m high Shanghai Tower (SHT) in Shanghai, together with the observation of $PM_1$ chemical composition. In section 2, we describe the measurement sites, instruments and analysis methods. In section 3, we discuss the general characteristics, seasonal variations and diurnal cycles of both SHT and SUR PM. Then, a conclusion is presented in section 4.

## 2 Experimental

### 2.1 Measurement site

Shanghai seats in the east of Yangtze River Delta region of China and is under the influence of northern subtropical monsoon. As mentioned before, the measurement site is located on a platform (~600 m) at the top of SHT (121.501°E, 31.236°N) in Lujiazui Finance and Trade Zone, a typical central business district with local emissions mainly from dense transportation. As the world's second highest construction that has been finished in the world, SHT stands out in the skyline (Figure 1). To compare the PM characteristics between SUR and SHT, SUR PM data and meteorological data were collected at Pudong Environmental Monitoring Center (PEMC) site and Pudong Meteorological Bureau (PMET) site respectively. PEMC site (121.534°E, 31.229°N, about 3.2 km east of SHT) deployed by Shanghai Environmental Monitoring Center belongs to the national air quality monitoring network, providing hourly concentrations of $PM_{2.5}$, $PM_{10}$, sulphur dioxide ($SO_2$), carbon monoxide (CO), ozone ($O_3$), and nitrogen dioxide ($NO_2$) for this study. Hourly meteorological measurements including 2 m air temperature, relative humidity (RH), 10 m horizontal wind speed and direction were obtained at PMET site (121.548°E, 31.222°N, about 4.7 km east of SHT) (Pan et al., 2019), which is a standard meteorological observatory managed by Shanghai Meteorological Bureau. Both PEMC site and PMET site are referred to as SUR site in the following discussion. All data are presented in Beijing Standard Time (BJT), which is 8 h ahead of Universal Time Coordinated (UTC).

### 2.2 Instrumentation

The Aerodyne quadrupole-type Aerosol Chemical Speciation Monitor (Q-ACSM) was equipped at SHT to analyze non-refractory $PM_1$ (NR-$PM_1$) (Canagaratna et al., 2007) chemical components, including sulfate ($SO_4$), nitrate ($NO_3$), ammonium ($NH_4$), chloride (Chl) and organics (Org) (Ng et al., 2011b), with a time resolution of ~15 minutes. The Q-ACSM was deployed for one year from April 17, 2019 to April 16, 2020. The particles greater than 2.5 μm were removed

through a $PM_{2.5}$ cyclone (Model URG-2000-30ED) in front of the sampling line. The particles were then dried with a nafion
dryer (Perma Pure, Model MD-700-36S-1) before passing through the ACSM inlet. Moreover, $PM_1$ and $PM_{2.5}$ mass
concentrations were collected simultaneously at a time resolution of 5 minutes by using the Thermo Scientific Model 5030
SHARP monitor. The nitrogen oxide ($NO$-$NO_2$-$NO_x$) and $SO_2$ data were collected by the Thermo Scientific Model 42i and
43i, respectively. Both the gas and aerosol analyzers are deployed in a cabin equipped with air condition (Figure 1),
providing continuous measurements with high reliability for this study.

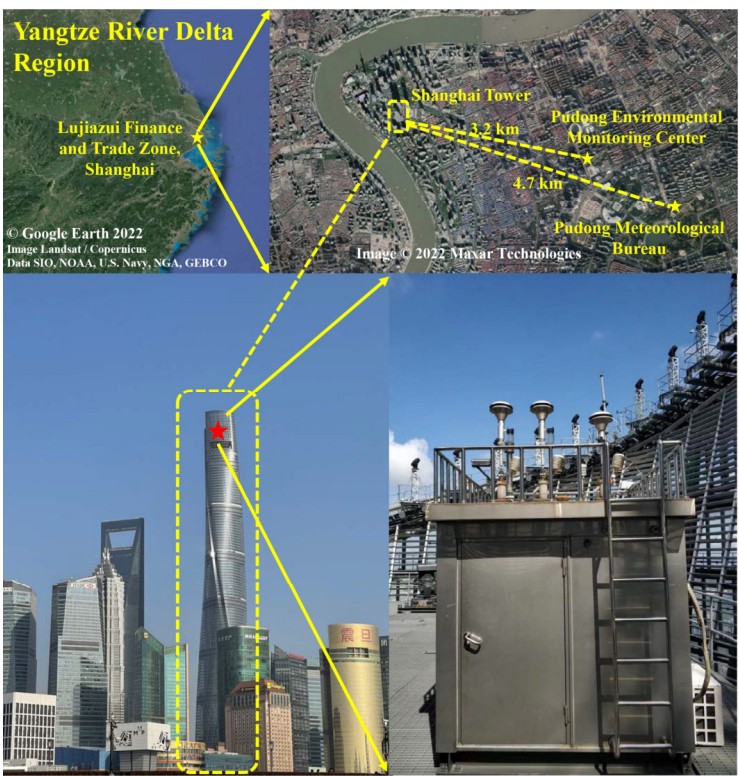

**Figure 1: The deployment of SHT site. The upper-left image gives a view of Yangtze River Delta Region (screenshot from Google Earth 2022 map data: Data SIO, NOAA, U.S. Navy, NGA, GEBCO. Image Landsat/Copernicus). The upper-right image shows the map of sample sites (Image © 2022 Maxar Technologies). The red star in the lower-left photo denotes the platform at the top of**
**SHT.**

## 2.3 ACSM data analysis

The ACSM chemical species concentrations were determined from the ion signals measured by aerosol mass spectrometer,
using the ACSM Local software (version 1.6.1.0, released on October, 2017) within Igor Pro (Wave Metrics, Inc., USA).
The relative ionization efficiency (RIE) values were determined as 7.99 and 0.81 for $NH_4$ and $SO_4$ through ionization
efficiency (IE) calibrations following the procedures brought by Ng et al. (2011b). RIE values were set as defaults for $NO_3$
(1.1), Org (1.4), and Chl (1.3). The value of collection efficiency (CE) was taken as 0.5 based on previous field studies in

China (Zhao et al., 2020a; Chen et al., 2015; Huang et al., 2012). The value of 0.5 is reasonable as aerosol particles were dried, and the mass fraction of ammonium nitrate (29%) was below the threshold value (40 %) that affects CE (Middlebrook et al., 2012). Besides, the ratio of measured $NH_4$ and predicted $NH_4$ was 0.78, indicating that the particle acidity was weak,
and had little effects on CE. The composition dependent CE was investigated according to the algorithm brought up by Middlebrook et al. (2012) and resulted in no significant changes.

**2.4 Source apportionment**

The organics data were further examined by source apportionment using PMF (Canonaco et al., 2013) with an Igor-based source finder tool (SoFi version 6.G). Only $m/z$ lower than 120 were included in the source apportionment analysis. The
unconstrained two-factor situation (Figure 2) was chosen for following discussions. One factor was recognized as a mixture of primary OA (POA). The corresponding profile had hydrocarbon-like fragments ($C_nH_{2n-1}$ and $C_nH_{2n+1}$; particularly $m/z$ 27, 29, 41, 43, 55, 57, 67, and 71) as in HOA, higher ratio of $m/z$ 55 than $m/z$ 57 as in COA (cooking OA), and distinctive polycyclic aromatic hydrocarbons (PAHs) fragments as in CCOA (coal combustion OA) (Duan et al., 2019). The mixture of POA factors was also reported in previous ACSM studies (Sun et al., 2018). The other factor had an obvious OOA signature
with a profile of significantly higher contribution (28.8%) of $m/z$ 44 ($CO_2^+$) as compared with previous ambient AMS/ACSM datasets (Ng et al., 2011a), meaning that OOAs reaching SHT experienced sufficient chemical aging processes. After increasing the number of unconstrained factors (Figure S1), no extra meaningful factors were interpreted. We also tried performing PMF analysis separately for each season (Figure S2-5), the POA factors were mixed with OOA feature (prominent $m/z$ 44 signal) in 2-factor solutions for all four seasons. We did ME-2 analysis with a priori POA profile from the
unconstrained two-factor solution for the entire research period, two OOA factors could be identified as a MO-OOA (more oxidized OOA) and a LO-OOA (less oxidized OOA) (Figure S6-8). Then, we compared the mass concentrations of ME-2 OA factors with those of unconstrained factors. The Pearson correlation coefficients ($R^2$) between the two methods were 0.97 and 1.00 for POA and OOA, respectively. However, a portion of 22.3% of unconstrained POA mass further split into OOA in the ME-2 solution. As lack of simultaneous measurements of surface chemical components, the unconstrained 2-
factor solution of PMF was adopted in consideration of focusing on PM differences between SHT and SUR.

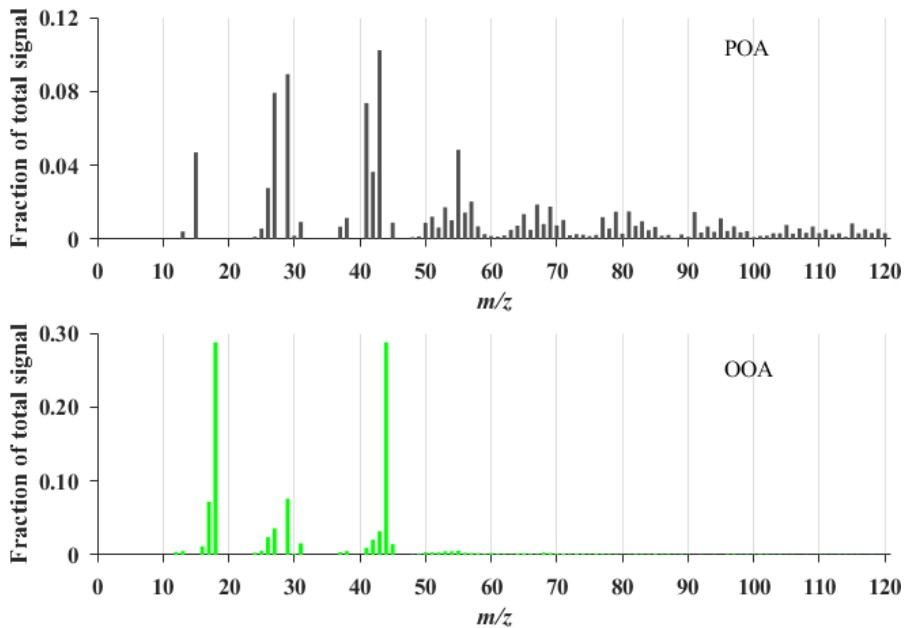

**Figure 2: Mass spectra profiles of OA factors for unconstrained two-factor situation.**

### 2.5 PBL height analysis

To diagnose the relative location of the SHT site to PBL, the seasonal and diurnal changes of PBL height (PBLH) were introduced. We obtained PBLH at SHT from the nearest ERA5 gridded reanalysis data (Hersbach et al., 2020) (https://cds.climate.copernicus.eu/cdsapp#!/dataset/reanalysis-era5-single-levels?tab=form, last access: 27 November 2022). The ERA-PBLH is calculated utilizing a bulk Richardson method, which was widely used for both convective and stable boundary layers (Kim, 2022). According to Wang et al. (2018), the ERA data tend to overestimate PBLH at nighttime, but underestimate PBLH during daytime in Eastern China by comparing with PBLH calculated from radiosonde sounding data. Overall, the reanalysis data can capture the diurnal and seasonal cycle of PBL structure.

As shown in Figure 3, the autumn found the highest PBLH for its prevailing synoptic of the continental high pressure (characterized as weak winds, strong solar radiation, and dry weather), favorable for the PBL development. PBLH in four seasons presented similar diurnal variations. The PBL started to develop at 06:00-08:00 before reaching a daily top at 13:00-14:00, and then decreased until stabilizing after sunset (18:00-19:00). However, the summertime PBL had the longest development period (06:00-19:00), while the wintertime PBL had the shortest (08:00-18:00). At nighttime, the observatory at SHT generally stood on top of stable PBL despite the deviations. Whereas the time PBL top reaching SHT site varied during the day. Nevertheless, the PBL had contact with SHT top even for the lower bound of deviation, indicating inevitable mass exchanges between SHT and SUR during the daytime.

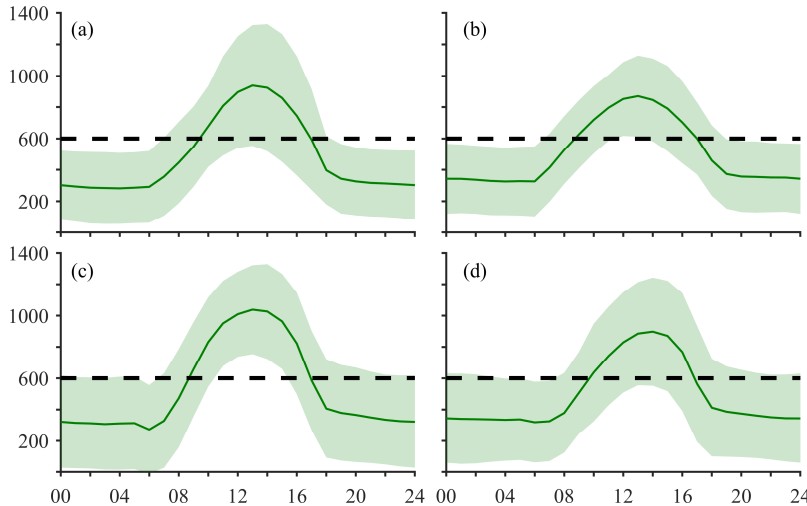

**Figure 3: Diurnal variations of the reanalysis PBLH in spring (a), summer (b), autumn (c), and winter (d) at the grid box where the Shanghai Tower (SHT) site is in. The solid line represents the mean value, and the shaded area stands for the standard deviation. The dash lines represent the altitude (~600 m) of the SHT site.**

## 3 Results and discussions

First, it should be noted that we omitted the PM originating from transport outside Shanghai throughout the discussion. However, the seasonal winds induced by Asia monsoon are quite different in upstream (ocean or land, mostly natural or anthropogenic in background) and could impact much at SHT than on the surface. We analyzed the transport pathway at the height of 100 m and 600 m in each season, using 72 h back trajectory from HYbrid Single Particle Lagrangian Integrated Trajectory (HYSPLIT) model. Though the two heights had similar tracks (Figure S9), the small departures might lead to different source origins. This factor should be explored in future studies. Second, PBLH is crucial for the vertical structure analysis, and direct observations of PBLH are in need to give precise view of the boundary layer processes.

### 3.1 Overview of PM levels and chemical compositions

### 3.1.1 PM$_{2.5}$ and PM$_1$ concentrations

The averaged PM$_{2.5}$ concentration (Table 1) at SHT during the observation period was 25.5±17.7 µg m$^{-3}$, about 16.4% lower than that (30.5±20.7 µg m$^{-3}$) at SUR. In which, the PM$_{2.5}$ measurements at SHT were generally consistent with those obtained from sensor-based instruments at the same platform reported by Hao et al. (2022) but only covered the period from June to November of 2019. The averaged PM$_1$ concentration at SHT was 17.3±11.7 µg m$^{-3}$ (68% of PM$_{2.5}$), also lower than reported surface PM$_1$ measurements in Shanghai (e.g., Qiao et al., 2015; Zhou et al., 2018a). The PM$_1$/PM$_{2.5}$ ratio at SHT was comparable with those reported by Qiao et al. (2016) and close to 0.69 presented by Zhou et al. (2018a), suggesting the

main contribution of $PM_1$ to $PM_{2.5}$. $R^2$ between SHT and SUR was 0.61 for hourly $PM_{2.5}$ concentrations. By contrast, $R^2$ of $PM_{2.5}$ between PEMC and other surface sites in Pudong district were all higher than 0.89, indicating more significant inhomogeneity of PM distribution in vertical than in horizontal. The relatively lower $R^2$ in vertical direction suggests distinct origins, transformations and fates of PM at upper PBL which need to be explored.

### 3.1.2 Chemical compositions

The averaged concentration of $NR-PM_1$ measured by ACSM was $16.4\pm3.6$ µg m$^{-3}$, which was little lower than the $PM_1$ concentration from SHARP 5030, indicating small black carbon existence. Among $NR-PM_1$, the averaged concentrations of chemical species were $3.4\pm2.2$ µg m$^{-3}$ (20.9%) for $SO_4$, $4.7\pm5.3$ µg m$^{-3}$ (28.6%) for $NO_3$, $2.1\pm1.7$ µg m$^{-3}$ (12.9%) for $NH_4$, $1.9\pm1.5$ µg m$^{-3}$ (11.4%) for POA, $4.0\pm2.8$ µg m$^{-3}$ (24.6%) for OOA, $0.3\pm0.2$ µg m$^{-3}$ (1.6%) for Chl. In general, OA, $NO_3$ and $SO_4$ were the first three leading contributors to $NR-PM_1$, consistent with the ACSM measurements at 260m Beijing tower (Chen et al., 2015). Similar with previous surface observations in Shanghai (e.g., Zhu et al., 2021; Zhao et al., 2020a), the OA dominated $PM_1$ with 35.8% contribution at SHT, in which, POA and OOA comprised 31.5% and 68.5% respectively, and the fractions were very close to the 260m observations (39% and 61%) in Beijing (Chen et al., 2015). The $NO_3$ at SHT (28.6%) had larger contribution than documented surface measurements (about 1-26%, summarized in Table S1) (Zhao et al., 2020a; Zhu et al., 2021). We found significantly higher $NO_3$ mass fractions in spring and winter at SHT than previous surface studies despite sampling sites, instruments, and years. In summer, the fraction at a rural site (Zhao et al., 2020) was found lower but close to this study. According to Cui et al. (2022), the proportion of $NO_3$ was as high as 26.0% in late autumn of 2018. For a similar period (November) in 2019, the ratio at SHT was 27.2%. We also gathered surface water-soluble $NO_3$ from MARGA (Monitor for AeRosols and Gases in ambient Air) observations at PEMC site for the exact same period of this study, further supporting the higher portion of $NO_3$ at SHT. The $R^2$ between $NR-PM_1$ and SHARP $PM_1$ was 0.82, indicating consistency between measurements of ACSM and SHARP 5030. Both $NR-PM_1$ and SHARP $PM_1$ showed a decreasing frequency in mass concentration (Figure S10), while a positive-skewed distribution was found for $PM_1$ at SUR in previous study in Shanghai (Zhao et al., 2020a). These results can be attributed to that SHT is far away from emission sources, and influenced by lower PM background concentration than SUR.

### 3.1.3 Meteorological elements

The observatory at SHT is close to the top of PBL, observed airmass was less affected by direct exchange of heat and moisture from surface. The meteorology at SHT presented lower temperature and less relative humidity (RH) than those at SUR. For example, the mean temperature at SHT was 3-4 °C lower than SUR in different seasons. In terms of RH at SHT, it was nearly consistent with that at SUR in summer and autumn, while about 5-10% lower in spring and winter. Both temperature and RH at SHT showed consistent seasonal variations with those at SUR.

The differences between maximum and minimum temperature at SHT were greatest (5.5 °C) in Spring, and smallest (4.1 °C) in Autumn (shown in Table 1). In comparison, the daily ranges of temperature at SUR were greater than SHT in all seasons, with largest range (8.8 °C) in Spring, and smallest (6.9 °C) in Summer. As the daily maximum (minimum) temperature always shows around noontime (midnight), greater temperature differences between SHT and SUR were presumed during daytime than those during nighttime. Accordingly, temperature-sensitive particle formations, for example gas-to-particle partitions were expected to have great differences between two altitudes during the daytime. The daily maximum RH at SHT were lower than SUR in all seasons, yet the daily minimum RH at SHT were higher than SUR. Unlike the temperature, the daily minimum (maximum) RH can always be found during daytime (nighttime). Thus, the higher daytime and lower nighttime RH were expected at SHT than those at SUR, leading to higher daytime and lower nighttime chemical productions from potential heterogeneous reactions at SHT.

**Table 1: The seasonal and annual averaged concentrations of aerosol species (μg m$^{-3}$) and meteorological parameters. The "dmean", "dmax" and "dmin" mean the daily average, maximum and minimum.**

| | | Spring | Summer | Autumn | Winter | Annual |
|---|---|---|---|---|---|---|
| Aerosol Species (μg m$^{-3}$) | | | | | | |
| SHT | PM$_1$ | 18.6±11.3 | 16.7±10.8 | 14.8±8.5 | 19.4±14.8 | 17.3±11.7 |
| | PM$_{2.5}$ | 25.5±14.2 | 22.4±13.0 | 22.3±13.5 | 31.4±24.7 | 25.5±17.7 |
| | SO$_4$ | 3.0±1.9 | 4.2±2.2 | 3.1±1.8 | 3.3±2.4 | 3.4±2.2 |
| | NO$_3$ | 4.8±4.8 | 3.3±3.2 | 3.4±2.9 | 7.2±7.6 | 4.7±5.3 |
| | NH$_4$ | 2.0±1.5 | 1.9±1.3 | 1.9±1.1 | 2.6±2.3 | 2.1±1.7 |
| | Chl | 0.2±0.2 | 0.1±0.1 | 0.3±0.2 | 0.4±0.3 | 0.3±0.2 |
| | OA | 6.1±3.8 | 6.6±5.2 | 5.0±2.9 | 5.8±4.1 | 5.9±4.2 |
| | POA | 1.9±1.4 | 2.4±2.1 | 1.5±1.0 | 1.7±1.2 | 1.9±1.5 |
| | OOA | 4.1±2.6 | 4.2±3.2 | 3.5±2.1 | 4.2±3.0 | 4.0±2.8 |
| SUR | PM$_{2.5}$ | 29.0±15.8 | 24.7±12.9 | 24.3±14.3 | 43.7±29.1 | 30.5±20.7 |
| Meteorological parameters | | | | | | |
| SHT | T-mean (°C) | 13.3±5.3 | 22.8±3.1 | 15.9±4.7 | 5.9±3.7 | 14.5±7.4 |
| | T-dmax (°C) | 16.2±5.7 | 25.4±3.3 | 18.0±4.9 | 8.2±4.2 | 17.0±7.7 |
| | T-dmin (°C) | 10.7±5.3 | 20.7±3.2 | 13.9±4.7 | 3.6±3.3 | 12.2±7.5 |
| | RH-mean (%) | 61.1±21.5 | 79.6±9.0 | 74.9±11.6 | 72.1±15.4 | 71.9±16.6 |
| | RH-dmax (%) | 74.9±19.2 | 88.9±4.8 | 84.8±9.0 | 82.3±11.5 | 82.7±13.3 |
| | RH-dmin (%) | 46.3±23.6 | 67.4±13.6 | 61.9±14.4 | 59.6±19.6 | 58.8±19.8 |
| SUR | T-mean (°C) | 16.2±4.6 | 26.5±3.1 | 19.7±4.8 | 8.6±3.0 | 17.7±7.6 |
| | T-dmax (°C) | 20.7±5.5 | 30.3±3.6 | 23.7±5.0 | 12.3±3.9 | 21.8±7.9 |
| | T-dmin (°C) | 11.9±4.6 | 23.4±3.2 | 16.1±5.3 | 5.2±3.3 | 14.2±7.8 |
| | RH-mean (%) | 71.0±15.1 | 82.8±8.1 | 76.7±10.5 | 77.5±13.4 | 77.0±12.7 |
| | RH-dmax (%) | 94.4±7.3 | 97.3±4.0 | 95.1±7.8 | 94.8±8.9 | 95.4±7.3 |
| | RH-dmin (%) | 46.1±23.1 | 63.1±13.1 | 53.3±16.0 | 55.0±21.5 | 54.4±19.7 |

 **3.2 Seasonal changes**

### 3.2.1 Monthly variations of PM$_{2.5}$ at SHT and SUR

As shown in Figure 4a, the monthly variations of PM$_{2.5}$ at SHT and SUR were generally consistent, higher in winter and lower in late summer to early autumn. The highest monthly PM$_{2.5}$ concentrations at SHT (36.9 μg m$^{-3}$) and SUR (52.6 μg m$^{-3}$) both took place in January, while the minimums (17.5 μg m$^{-3}$ for SHT, and 18.8 μg m$^{-3}$ for SUR) appeared in September.

The significant monthly change of PM$_{2.5}$ is resulted from the distinct primary emissions (aerosol and its gas precursors), chemical transformation, transports and diffusions as well as wet removals in different seasons. In winter, more pronounced transport and much shallower PBL are conductive to PM$_{2.5}$ accumulation near surface, resulting in higher SUR PM$_{2.5}$ loadings in Shanghai. It is noting that daytime PBL developments usually carry aerosols from surface to high altitude by turbulence, exerting opposite impacts on PM$_{2.5}$ variations near surface and at upper altitudes. As a result, local emissions and

regional transports were expected to be responsible for the similar patterns of PM$_{2.5}$ monthly variation at SHT and SUR, rather than PBL changes.

### 3.2.2 Monthly anomaly of PM$_{2.5}$ at SHT and SUR

The anomaly was defined as the monthly deviation from annual average. By calculating the anomaly, we intended to find monthly changes relative to the whole year. The comparison of SHT and SUR PM anomalies allows us to see the

235 consistency of monthly features at two altitudes. For both SHT and SUR, negative PM$_{2.5}$ anomalies were found in most months, because of significant positive anomalies in January and December (Figure 4b). The PM$_{2.5}$ concentrations in January and December were 11.6 μg m$^{-3}$ and 22.2 μg m$^{-3}$ higher than the annual averages at SHT and SUR, respectively. Therefore, more stringent attentions should be paid to PM mitigation during these two months. The monthly changes of PM$_{2.5}$ anomalies presented very similar patterns at SHT and SUR, with largest positive anomaly in January, highest negative one in

August and September. However, the SHT anomaly oscillated more flatly than SUR, for example, the relative anomaly (PM anomaly divided by yearly-averaged value) during December to January (39.7%) and August to September (-20.2%) was much lower than those at SUR (62.8% and -32.2%). It could be partially explained by weaker influences from surface emissions and air pollutants at higher altitudes, due to nighttime isolations and shallow PBL in winter discussed in following section. It was noting that PM$_{2.5}$ anomalies were opposite in February and October compared with those at SHT and SUR,

indicating different leading roles on PM$_{2.5}$ budgets between surface and mid-upper PBL.

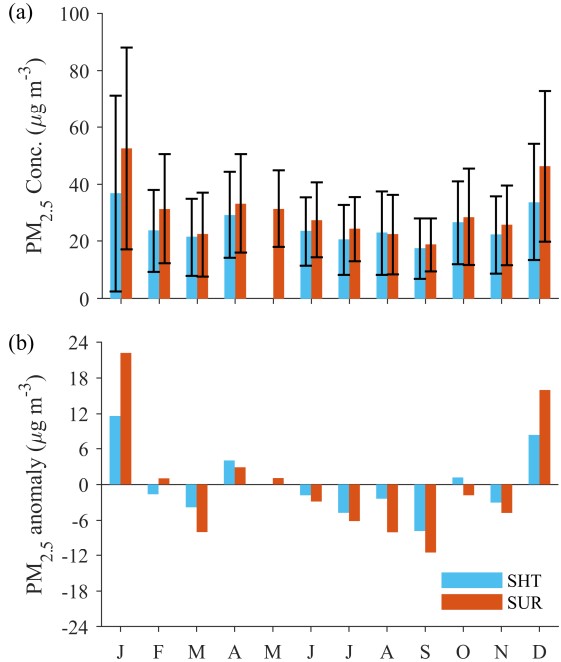

**Figure 4: Monthly variations of (a) PM$_{2.5}$ concentrations (µg m$^{-3}$) at SHT and SUR, (b) PM$_{2.5}$ anomalies (µg m$^{-3}$) at SHT and SUR. The monthly-averaged PM$_{2.5}$ in May at SHT is not presented because of low data collection efficiency (36%).**

### 3.2.3 Relative changes between monthly-averaged PM at SHT and SUR

Since lower PM$_{2.5}$ concentrations were observed at SHT compared with SUR, the relative percentage changes (RPC) ((PM$_{2.5,}$ $_{SHT}$ − PM$_{2.5, SUR}$)/PM$_{2.5, SUR}$*100%) in Figure 5 were calculated to quantify their discrepancy. The RPC (Figure 5a) exhibited the lowest value (-27.2%) in winter (December, January and February), and generally consistent values (-8.0%, -9.2%, and -9.1% for spring, summer, and autumn) in the other seasons. It was noting that PM$_{2.5}$ concentration observed at SHT was slightly higher (2.4%) than SUR in August. Given that SHT was farther from the direct emission sources than SUR, the PM$_{2.5}$ at SHT tended to have lower concentration than SUR as in the other months despite vertical mixing during the daytime. Thus, the higher PM$_{2.5}$ at SHT in August indicated extra aerosol productions at mid-upper PBL.

The exchange of air pollutants between SHT and SUR only exits in daytime due to turbulent mixing. Such mixing process between SHT and SUR would terminate at night due to stable stratification. It could be presumed that nighttime PM$_{2.5}$ observed at SHT (600 m) was independent from that at SUR. Therefore, it is necessary to compare the daytime and nighttime PM$_{2.5}$ separately between SHT and SUR. In Figure 5b, nighttime PM$_{2.5}$ concentrations observed at SHT were consistently lower than SUR, with RPC ranging from -20% to -38%. In addition, nighttime RPC presented very weak monthly variations, suggesting that PM$_{2.5}$ observed at the height of SHT was mostly isolated from both air pollutants and primary emissions near surface at night. In comparison, daytime RPC presented strong monthly variabilities. Different from

the negative RPCs of daily PM2.5 presented in Figure 5a, daytime PM2.5 concentrations at SHT were 20-40% higher than those at SUR from June to October. As is known, turbulence induces the vertical mixing of PM2.5 to eliminate the PM2.5 gradient within the entire PBL. Since there are no direct primary sources at SHT, the higher PM2.5 measurements mean
additional physical or chemical origins existed at this height. Thus, the daytime RPC modulated the seasonal signature of the total RPC, presenting the lowest RPC in winter and implying the necessity of looking into PM diurnal changes.

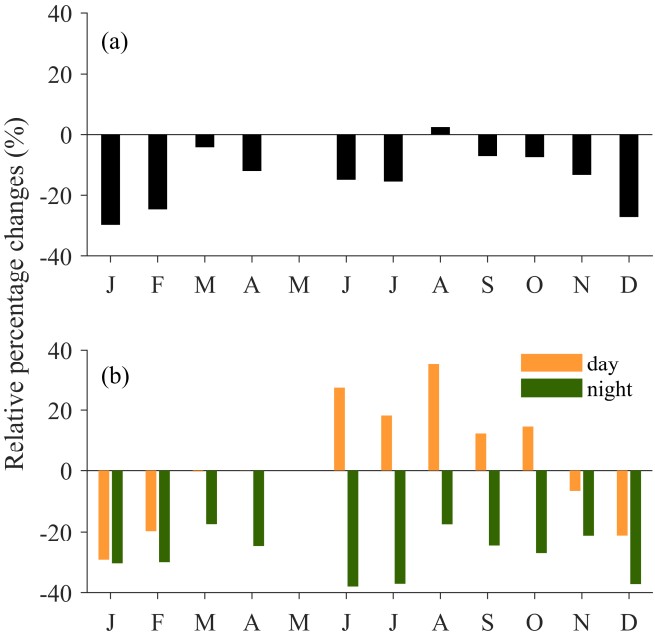

**Figure 5: Monthly variations of the relative percentage changes (%) between SHT and SUR for (a) all time, (b)**
**daytime (08:00-19:00) and nighttime (20:00-07:00). The relative percentage changes are calculated as (PM$_{2.5, \text{SHT}}$ − PM$_{2.5, \text{SUR}}$)/PM$_{2.5, \text{SUR}}$*100%.**

### 3.2.4 Monthly variations of chemical species at SHT

As shown in Figure 6, the NR-PM$_1$ at SHT presented consistent variability and slight departures with SHARP PM$_1$, providing well insight to investigate the seasonal contributions of chemical composition. OA, NO$_3$ and SO$_4$ occupied the
280 three leading contributions to NR-PM$_1$ at SHT, accounting for 36%, 28.6% and 20.9% respectively. Their proportions in NR-PM$_1$ presented distinct seasonal variations. In general, SO$_4$ and OA occupied higher fractions in summer, and lower proportions in winter. For example, SO$_4$ had largest portion of 26.1% in NR-PM$_1$ in summer, while the lowest of 17.1% in winter. Similarly, OA made up the most proportion of 39.5% in summer, and the least of 29.9% in winter. Both OOA and POA showed consistent seasonal contributions to NR-PM$_1$ with OA. Larger fractions of OOA and SO$_4$ in NR-PM$_1$ exhibited
in summer were attributed to relative stronger oxidation capacity and higher moisture conductive to both gas and aqueous transformations, which were also observed by other mass spectrometer studies near surface (Dai et al., 2019; Hu et al.,

2016). Different from OOA, HOA was mainly emitted by vehicles especially in cold months. Some documents found that HOA dominated the organics in wintertime Beijing due to enhanced primary emissions from heating season (Duan et al., 2020; Zhang et al., 2013; Zhou et al., 2019). Zhu et al. (2021) also reported a slight decrease of HOA fraction in summer than other seasons in urban Shanghai. In this study, the POA observed at SHT comprised highest fraction of 14.6% in NR-$PM_1$ and 35.8% in organics in summer. In comparison, the ratios were 8.7% and 28.3% in winter, indicating that POA transport and mixing process were significantly inhabited in winter than in summer. It was expected that $NO_3$ presented opposite seasonal pattern to $SO_4$ and OA. The fraction was found highest (37%) in winter and lowest (21%) in summer, because of the temperature-dependent gas-particle partition. As for $NH_4$, its seasonal variation was found insignificant.

The daytime and nighttime mass fractions were also shown in Figure 6. As results of vertical mixing, the larger portions of primary species (POA and Chl) during daytime were notable, especially for summer and autumn. The changes of OOA, $NO_3$, and $NH_4$ were slight, with increase of OOA and $NH_4$, but decrease of $NO_3$ from nighttime to daytime. Accordingly, $SO_4$ saw lower fraction in NR-$PM_1$ during the daytime. More diurnal features of NR-$PM_1$ can be found in section 3.3.4.

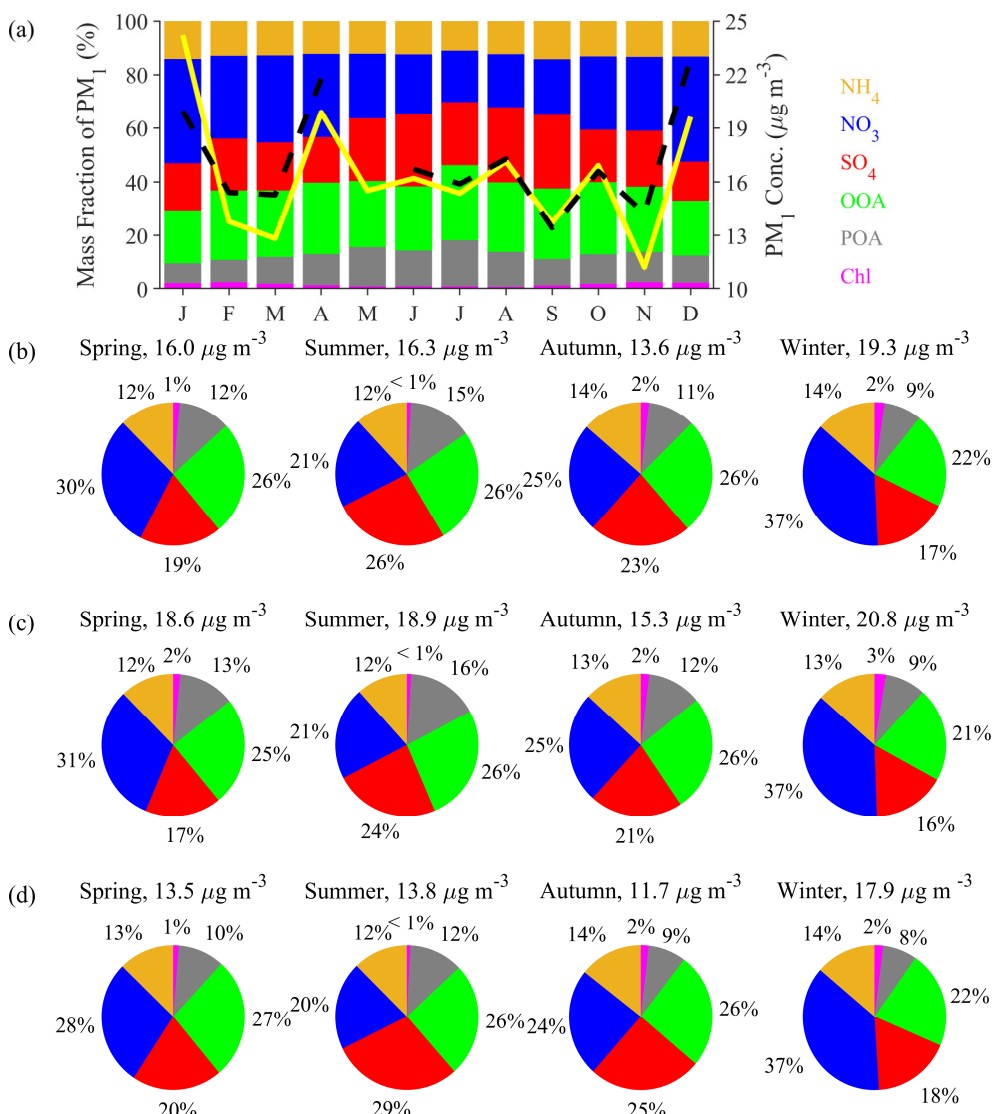

**Figure 6: The monthly averaged (a) and seasonal averaged (b-d) mass fractions (%) of NR-PM₁ at SHT. The mass fractions (%) are calculated based on all (b), daytime (c), and nighttime (d) data. The monthly averaged mass concentrations (µg m⁻³) of NR-PM₁ are also shown. The solid and dashed line represent SHARP PM₁ and NR-PM₁, respectively.**

## 3.3 Diurnal changes

### 3.3.1 Diurnal variations of PM₂.₅ at SHT and SUR

The air pollutants observed at mid-upper PBL have different origins from those near surface. They are effectively connected in daytime by turbulence, while absolutely isolated at night. As a result, the diurnal variations of air pollutants at bottom and upper PBL need to be investigated by synergetic observations. In this section, the diurnal characteristics of PM₂.₅ and

chemical compositions at SHT are displayed and compared with SUR to discuss the related dynamical and chemical processes at mid-upper PBL.

The mean diurnal variations of $PM_{2.5}$ observed at SHT and SUR are displayed in Figure 7, exhibiting different patterns in all seasons. The $PM_{2.5}$ diurnal cycle near surface have been fully documented by previous studies, driven by primary emission, PBL evolution and chemical transformation etc. In general, there existed a clear morning peak of $PM_{2.5}$ concentration around 6:00-7:00 in all seasons, due to substantial air pollutants were released from rush transportations and accumulated in shallow PBL. After sunrise, PBL gradually develops by turbulence which transports $PM_{2.5}$ from surface to high latitude. As a result, $PM_{2.5}$ concentration near surface decreased until early afternoon, then increased again after sunset due to depressed PBL and kept stable at night. However, the $PM_{2.5}$ concentrations observed at SUR presented stable or slight enhancement during the PBL developing period (10:00-15:00) in summer and autumn, differing from the clear $PM_{2.5}$ descent during the same period in spring and winter. It was also reported by Pan et al. (2019), they found the significant $PM_{2.5}$ enhancement around noontime in summer in Shanghai downtown, explained by that rapid production of secondary aerosols related to strong photochemistry in summer offsetting the aerosol loss by vertical mixing.

The $PM_{2.5}$ diurnal variation observed at SHT presented similar unimodal pattern in all seasons. Peak $PM_{2.5}$ concentration was observed around midnoon (12:00) as a result of the rapid increasing of $PM_{2.5}$ concentration since early morning, and notable decline in the afternoon. The early morning $PM_{2.5}$ enhancement at SHT was presumed to be resulted from the PBL development, transporting higher concentration of aerosols from surface to upper layer. Accordingly, $PM_{2.5}$ reduction at SUR was observed simultaneously in Figure 7 due to vertical mixing. It was interesting to note that higher $PM_{2.5}$ at SHT was observed than SUR around noontime (10:00 to 15:00) in spring, summer and autumn, which could not be attributed to vertical mixing. According to the turbulent theory, materials including heat, moisture and air pollutants in PBL are supposed to be mixed homogeneously by vertical mixing. Therefore, when $PM_{2.5}$ concentration at SHT exceed that at SUR before noontime in Figure 7, turbulence would mix the higher loadings of aerosol from upper PBL to surface. It could be also demonstrated by more significant $PM_{2.5}$ anti-correlations between SUR and SHT during morning to early noontime (6:00 to 10:00) when observed $PM_{2.5}$ at SHT was lower than SUR. Therefore, there must exist other processes responsible for the higher $PM_{2.5}$ concentrations appeared around noontime at SHT. This was also observed by Hao et al. (2022) that the daytime $PM_{2.5}$ concentration at 585m was larger than that at 25m by 10-15 $\mu g\ m^{-3}$. We supposed significant chemical formation of secondary aerosols as the dominant process for the occurrence of greater $PM_{2.5}$ levels at SHT around noontime, which were discussed in the following section.

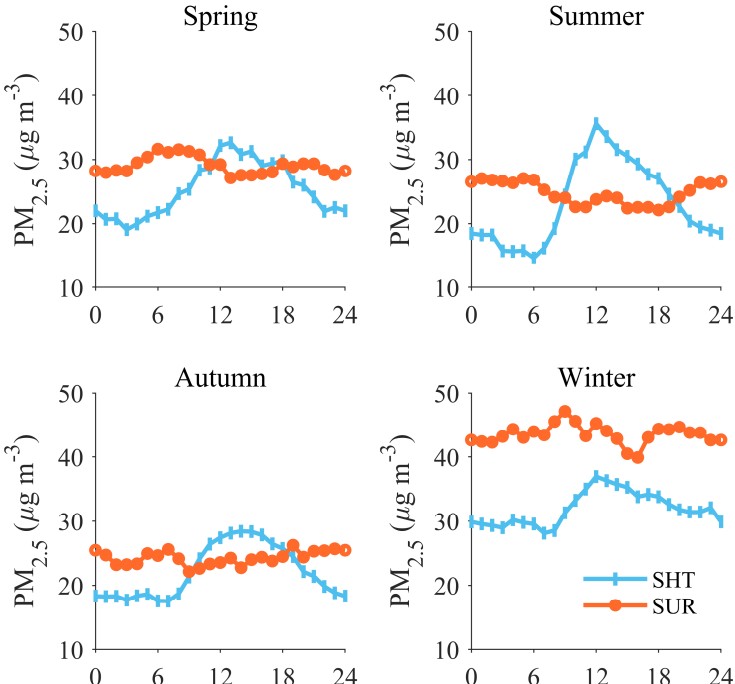

**Figure 7: Diurnal variations of PM$_{2.5}$ mass concentrations (μg m$^{-3}$) at SHT and SUR in four seasons.**

### 3.3.2 Discuss on the daytime aerosol production at SHT

As showed in Figure 7, the peak PM$_{2.5}$ concentration at SHT was highest (~40 μg m$^{-3}$) in summer, it could not origin from air pollutants near surface by vertical mixing because the PM$_{2.5}$ at SUR was relatively lower in summer. In addition, the PM$_{2.5}$ at SHT presented significant increasing rate even after its exceedance to SUR around 10:00, indicating that there must exist remarkable origins of aerosols at SHT, especially in summer. Therefore, chemical transformation from both gas and aqueous pathways were presumed to be the major process for promoting the aerosol productions at SHT, resulting in the significant

peak PM$_{2.5}$ concentration.

Chemical production of aerosol in daytime is mainly driven by gas phase and aqueous phase transformations from the gas precursors, including SO$_2$, NO$_2$ and VOCs. We further compared the gas species observed at SHT and SUR in Figure 8. It was found that SO$_2$ concentrations were low (<4 ppb) at both SHT and SUR, and SO$_2$ concentration at SUR was about 2

355    times higher than SHT. A slight increase (2.1-19.2%) of SO$_2$ at SHT during 8:00-12:00 could be found in four seasons. Similar as SO$_2$, lower NO$_2$ were observed at SHT than SUR. However, NO$_2$ was found well mixed around noontime in all seasons. NO$_2$ at SHT in different seasons rose by 21.8-61.4% from 8:00 to 12:00, when NO$_2$ were reduced at ground level, indicating the effects of vertical mixing. Therefore, secondary formation of inorganic aerosols could be expected at SHT with adequate gas precursors.

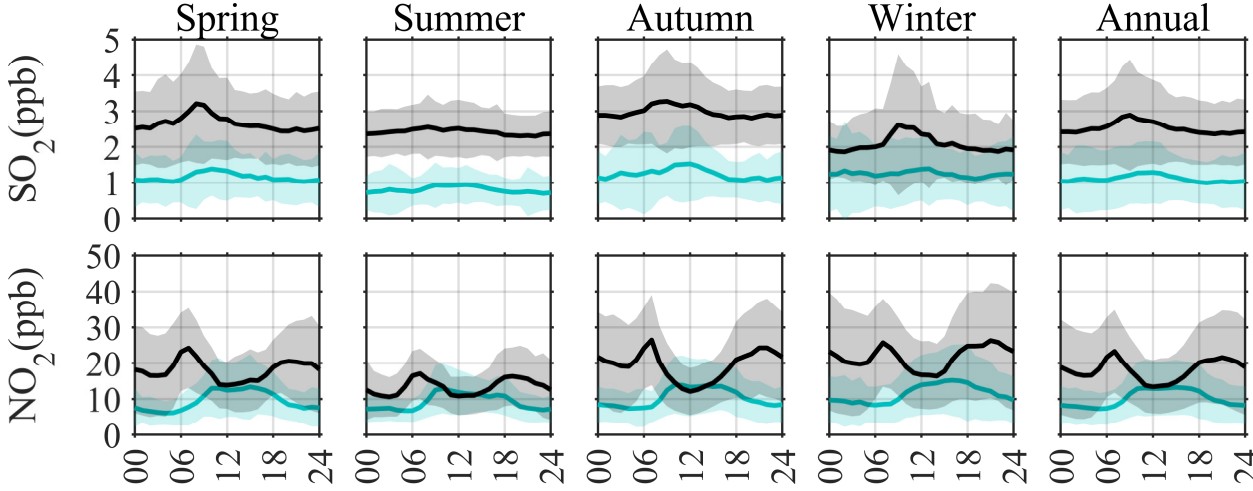

**Figure 8: Diurnal variations of SO$_2$ (ppb) and NO$_2$ (ppb) observed at SHT (blue line) and SUR (black line) in four seasons and the entire observation period. The line stands for mean value. The shaded area represents the standard deviation.**

Both gas and aqueous formation of secondary aerosols in daytime are greatly determined by atmospheric oxidants, such as OH, H$_2$O$_2$ radicals and O$_3$. In which OH is produced from the photodissociation of O$_3$ (clean atmosphere) or HONO/H$_2$O$_2$ (polluted atmosphere) in troposphere. So that solar radiation is vital for OH production. Apparently, stronger solar radiation could be expected at SHT than that near surface due to shorter optical range of sunlight and hardly attenuation from

buildings or vegetation, providing excellent photolysis capacity for OH production. We further examined the diurnal variations of sulfur oxidation ratio (SOR) and nitrogen oxidation ratio (NOR) at SHT in different seasons in Figure 9. SOR and NOR are indicators of the secondary formation of SO$_4$ and NO$_3$ (Zhang et al., 2020). The SOR and NOR are calculated as [SO$_4$]/([SO$_4$]+[SO$_2$]), and [NO$_3$]/([NO$_3$]+[NO$_2$]), respectively. [x] stands for the molar concentration of x. The SOR in summer (0.51) was significantly higher than those in the other seasons (0.36). The diurnal cycle of SOR was similar in each

season, with the highest SOR appeared around 19:00, when RH reached the peak of the day (Figure 10). However, the diurnal variations of NOR were not reproducible between the seasons and not as straightforward to interpret as those of SOR. NOR in spring and winter (0.15) were about 1.6 times larger than that in summer and autumn (0.09). In addition, NOR did not see notable decrease from 8:00 to 12:00, when NO$_2$ increased significantly, indicating that NO$_3$ formation at SHT was evident but not necessarily more efficient than SUR.

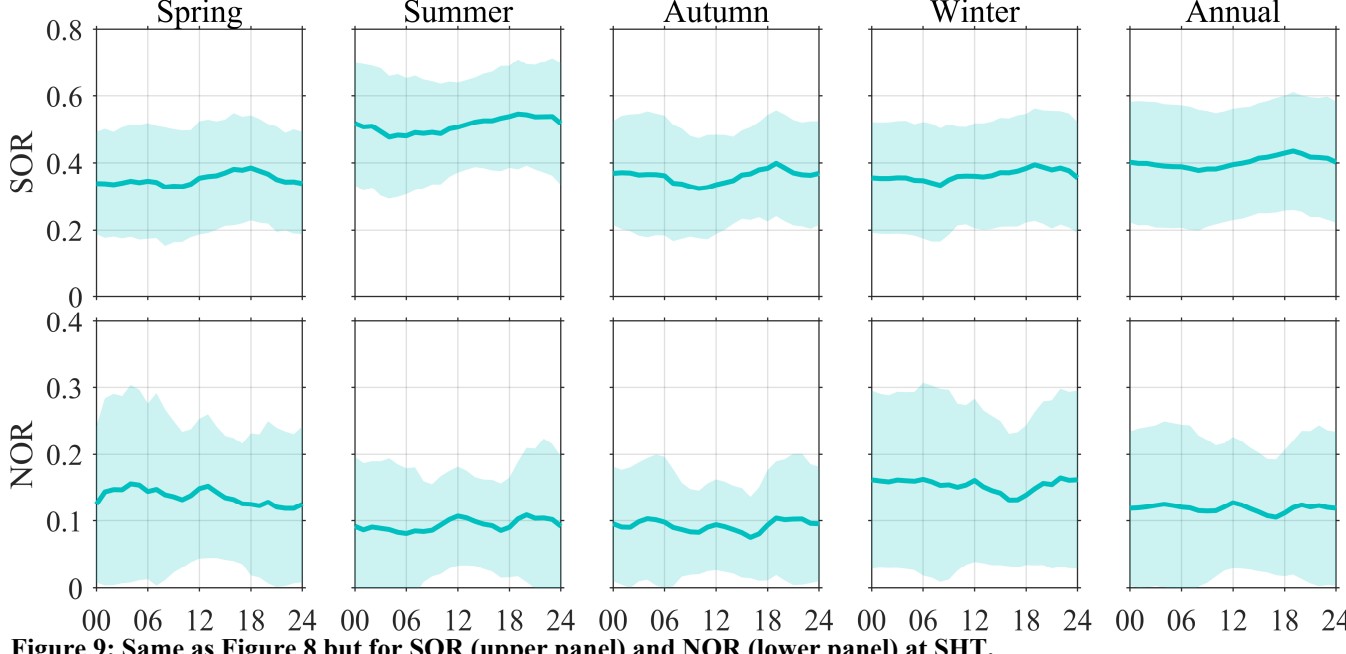

**Figure 9: Same as Figure 8 but for SOR (upper panel) and NOR (lower panel) at SHT.**

As was discussed in Figure 6, OOA, $NO_3$ and $SO_4$ were the major species in NR-$PM_1$. Their production and variation are closely related to meteorology, especially temperature and RH. For example, lower temperature was favorable for productions of $NO_3$ and some semi-volatile species in OA. Whereas higher RH played the important roles in the aqueous pathway of $SO_4$ formation. Recent studies also reported significant production of low-volatile OA by aqueous phase reactions (Chen et al., 2021). Considering the importance of meteorology, the diurnal variation of temperature and RH at SHT were displayed in Figure 10 and compared with SUR. In general, the SHT RH was found higher than SUR from 09:00 to 19:00 in all seasons. Therefore, the SHT atmosphere provided better conditions for the aqueous phase production of secondary aerosols during daytime. Similar to RH, the diurnal changes of air temperature were more visible for SUR. Overall, temperature at SHT was 4.4 °C lower than SUR during daytime, compared with 2.1 °C during nighttime. Furthermore, PM differences between SHT and SUR were found keen to both temperature and RH differences (Figure S11). On one hand, the lower temperature at SHT might correspond to a stronger vertical temperature gradient promoting vertical mixing of PM. What's more, the lower temperature was favorable for partition of $NO_3$ and semi-volatile organic species to particles.

According to above discussions, we suggest that the gas precursors, atmospheric oxidants and meteorology observed at SHT were all appropriate for aerosol formations through gas and aqueous pathway. As a result, efficient production of secondary aerosols could be expected at SHT in daytime, leading to the higher $PM_{2.5}$ concentration than SUR.

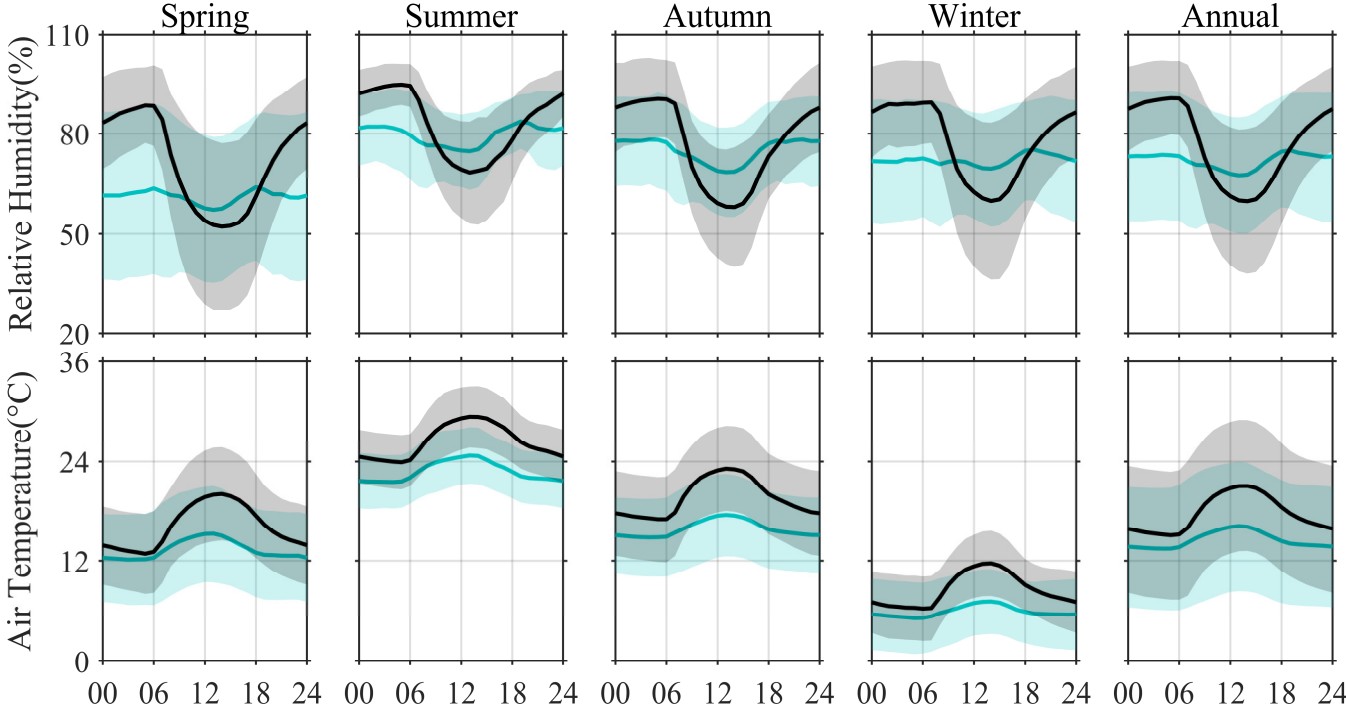

**Figure 10: Same as Figure 8 but for relative humidity (%, upper panel) and air temperature (°C, lower panel).**

### 3.3.3 Relative amplitudes of PM at SHT and SUR

Besides diurnal tendency, diurnal amplitude of $PM_{2.5}$ variations also presented clear distinctions between SHT and SUR. We introduced $(C_{max}-C_{min})/C_{mean}*100\%$ to estimate the relative amplitudes (Ramps) of atmospheric species. $C_{max}$, $C_{min}$, and $C_{mean}$ represent the maximum, minimum, and average of diurnal concentrations. The Ramps of $PM_{2.5}$ at SUR were 15.5%, 20.0%, 16.9% and 16.6% in spring, summer, autumn and winter, while those at SHT were 53.4%, 89.9%, 48.9%, and 27.7% respectively. Apparently, the diurnal $PM_{2.5}$ at SHT exhibited much larger Ramps than SUR, indicating more significant

amplitude. This was because nighttime $PM_{2.5}$ at SHT was much lower by isolating from surface. As a result, the nighttime $PM_{2.5}$ at SHT were 30-40% lower than that at SUR. While in daytime, the $PM_{2.5}$ departures between SHT and SUR were greatly narrowed to 10-20% due to turbulent mixing. In addition, chemical production still played important roles in higher Ramps at SHT. It could be demonstrated by the highest Ramps happened in summer at SHT due to the significant production of aerosols in daytime. In winter, the $PM_{2.5}$ enhancement during late morning to noontime was still observed at SHT, but

with weaker amplitude than those in other seasons. Since the $PM_{2.5}$ concentration at SHT during this period was continuously lower than SUR, vertical mixing made positive contributions to $PM_{2.5}$ concentration at SHT. It was supposed that the chemical production of aerosols at SHT was significantly inhibited due to weak solar radiation and few precursors

from vertical mixing in winter except for $NO_3$. While the mean $PM_{2.5}$ level was still highest in winter (Figure 7), resulted from the greater nighttime $PM_{2.5}$ level of 30 μg m$^{-3}$, significantly higher than those in other seasons (15-20 μg m$^{-3}$).

### 3.3.4 Diurnal variation of chemical compositions at SHT

The diurnal variations of chemical species at SHT were further examined in Figure 11. The major species, such as $NO_3$, $SO_4$ and OOA exhibited unimodal pattern similar as $PM_{2.5}$, with the peak concentration appearing around noontime. These were different from the 260m observations at Beijing reported by Chen et al. (2015), their peak concentrations of secondary species usually appeared at 20:00, indicating the distinct controlling process for $PM_{2.5}$ diurnal variation between lower and upper PBL. The pronounced increasing rate of $NO_3$ and OOA were observed during early morning (~8:00) to midnoon (~12:00) in spring, summer and autumn. For example, the increasing rates were estimated as 0.39 μg m$^{-3}$ h$^{-1}$ (9.3 % h$^{-1}$) and 0.29 μg m$^{-3}$ h$^{-1}$ (8.5 % h$^{-1}$) for $NO_3$ and OOA respectively, further demonstrating the significant chemical productions at SHT. $NO_3$ presented significant larger levels in winter and spring compared with other species. Both daytime and nighttime $NO_3$ accounted for the largest fractions of 36% in winter and 30% in spring in NR-$PM_1$. Zang et al. (2022) suggested that both heterogeneous hydrolysis of dinitrogen pentoxide ($N_2O_5$) and the gas-phase OH oxidation of $NO_2$ were the important pathways for nitrate formation in Shanghai. It was noting that nighttime $NO_3$ in winter was extremely high up to 6-7 μg m$^{-3}$, even exceeded daytime peak concentration in other seasons, indicating the most efficient heterogeneous production at winter night. In addition, $NO_3$ increasing rate in winter morning was estimated as 0.42 μg m$^{-3}$ h$^{-1}$, significantly higher than other species. It was attributed to the synergetic impacts of lower temperature favorable for the gas-to-aerosol partition, and higher $NO_2$ concentration promoting gas-phase $HNO_3$ productions. The enhanced level of $NO_3$ with altitude in PBL was also reported by other studies (Zhou et al., 2018b). OOA presented most significant diurnal variability in summer, suggesting that OOA formation was more sensitive to atmospheric oxidation. $SO_4$ formation in daytime is mainly driven by aqueous reactions. Therefore, higher $SO_4$ at SHT in summer was expected from plenty moisture and strong solar radiation accelerating its aqueous transformation, despite low $SO_2$ level, favorable diffusion, and wet scavenging condition of atmosphere in summer. However, $SO_4$ presented lower diurnal variabilities in all seasons compared with $NO_3$ and OOA. In general, distinct mass fractions of increased $PM_1$ (during 8:00-12:00) were observed with great contributions from $NO_3$ and organics in spring (80.9%), summer (85.4%) and autumn (83.0%). The notable differences of mass fractions between SHT and previously documented surface measurements indicate that the increased $PM_1$ at SHT were combined results of strong chemical production as well as vertical mixing.

The diurnal Ramps of chemical species in the NR-$PM_1$ were further estimated. It was clear that POA had the most significant diurnal amplitude, with the Ramps of 89.1%, 112.1%, 99.0%, and 59.4% in spring, summer, autumn and winter respectively. Similarly, another primary species, Chl also had comparable Ramps of 95.8%, 102.0%, 58.7%, and 66.4% in each season, suggesting the dominant impact of turbulence on the diurnal variations of primary species. In comparison, the secondary species exhibited much lower diurnal Ramps, with 49.7% for $NO_3$, 41.9% for OOA, 34.2% for $NH_4$, and 17.5% for $SO_4$. The

larger Ramps of primary species at SHT could be well understood, their nighttime concentrations were very low due to the isolation from surface emissions, while in daytime the concentrations increased due to vertical mixing. In comparison, the Ramps reduction of secondary species mostly resulted from the chemical formations at night. The pronounced formation of secondary aerosols through heterogeneous pathway at night has been widely observed near surface by many studies. The recent study from Zang et al. (2022) reported that heterogeneous pathway accounted for 68% of NO$_3$ production in winter in Shanghai. It could be found in Figure 11 that the nighttime concentrations of secondary species were significantly higher than primary ones nearly in all seasons, suggesting that there also existed active heterogeneous formation of aerosols at night at upper PBL level except summer.

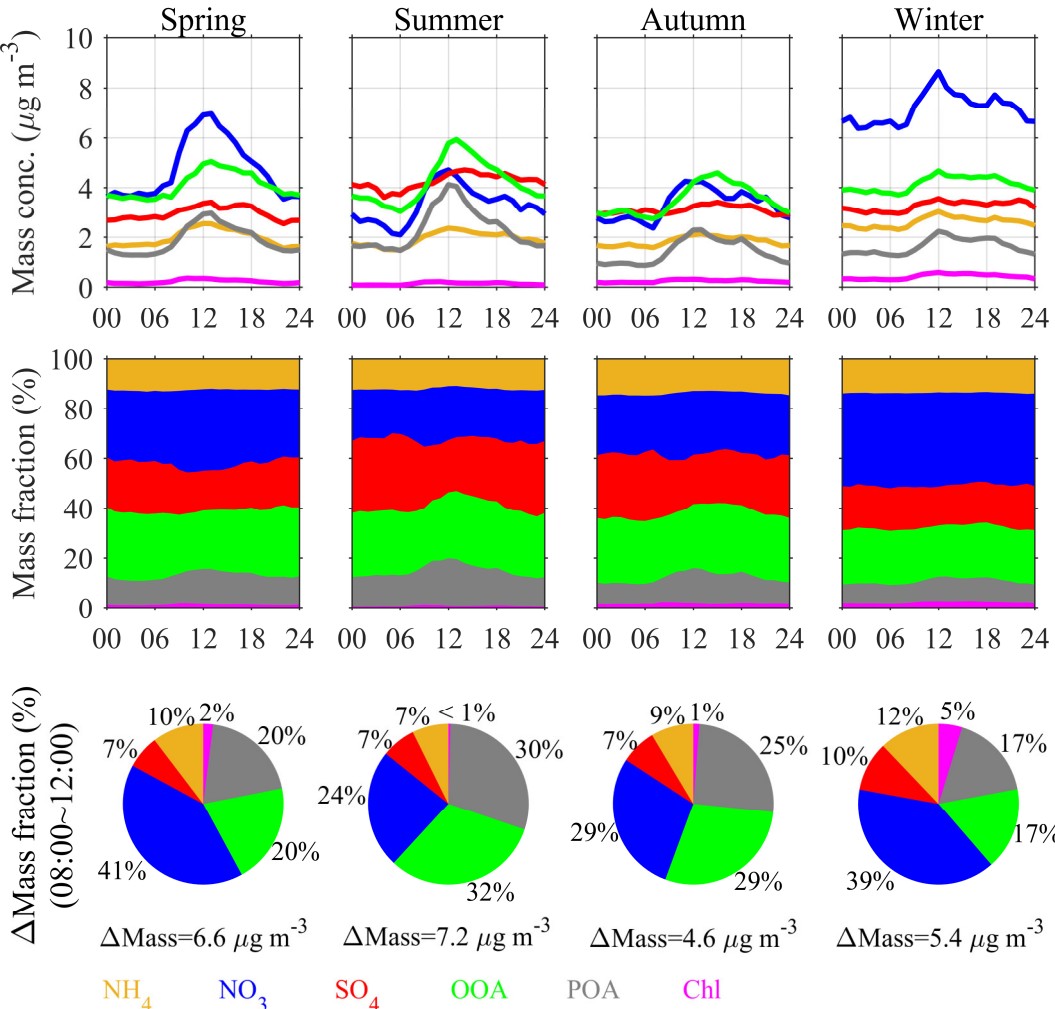

**Figure 11: Diurnal variations of NR-PM$_1$ mass concentration (µg m$^{-3}$) (upper panel) and mass fraction (%) (middle panel) at SHT in four seasons. Mass fractions (%) of increased NR-PM$_1$ during 08:00-12:00 (lower panel) in four seasons.**

## 4 Conclusions

This study presents one-year continuous observation of the fine PM mass concentrations and chemical compositions at the top of 632 m high Shanghai Tower from April 17, 2019 to April 16, 2020. The results show that SHT $PM_{2.5}$ concentration was 16.3% lower than SUR during the entire observation period. Through investigating the diurnal changes, we found uniformly lower nighttime $PM_{2.5}$ at SHT as results of isolations from surface emission. However, the daytime $PM_{2.5}$ presented significant monthly changes with unexpected higher concentrations than SUR from June to October. Other than surface, the SHT PM exhibited a consistent unimodal diurnal change in four seasons, with rapid increase of PM mass concentration starting from late morning, and a peak in the early afternoon. Combining the adequate precursors and lower temperature at SHT during daytime, we suggest strong chemical production of secondary species from both photochemical reactions and gas-to-particle partitioning at mid-upper PBL. Moreover, the averaged mass fraction of increased PM (during 8:00-12:00) at SHT revealed notably high proportions of $NO_3$ and organics, further demonstrating the contributions from chemical production as well as vertical mixing. In addition, we found high $NO_3$ concentration at SHT for both daytime and nighttime winter, implying efficient gas-phase and heterogeneous formation.

## Data availability

Hourly data used in this study are deposited at National Earth Observational Data Center (https://chinageoss.cn/datasharing/datasetDetails/630094ef42544e709be88207, last access: 27 November 2022), which provides open access to its data. Raw data of ACSM are archived at Shanghai Key Laboratory of Meteorology and Health, and are available on request by contacting the corresponding author.

## Author contributions

JMX designed the experiments and the research. CQY, LP, WG, YG, QF and FY provided experimental assistance and the analytical method. CQY and JMX analyzed the data and performed research. All authors commented on the manuscript.

## Competing interests

The authors declare that they have no conflict of interest.

# Acknowledgements

This research has been supported by Natural Science Foundation of Shanghai (grant no. 22ZR1467500), and the National Natural Science Foundation of China (grant no. 41605105).

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
