# Peer review of "Characteristics of fine particle matters at the top of Shanghai Tower"

_EGUsphere, 2022_

## Author Comment (AC1)

Dear Editor,

We appreciate the prompt reviews and would like to thank the two reviewers for insightful comments and suggestions on our manuscript entitled "Characteristics of fine particle matters at the top of Shanghai Tower" (MS No.: egusphere-2022-782). We have carefully considered all comments and suggestions. Listed below are our point-by-point responses to all comments and suggestions of this reviewer (Reviewer's points in black, our responses in blue).

**Anonymous Referee #1**

The author conducted chemical composition measurement at high altitude, expanding our understanding on aerosol chemistry at mid-upper PBL. I suggest major revision for the manuscript prior to be finally published in ACP.

**Response:**

We sincerely thank the reviewer for the valuable comments. These comments have been carefully addressed during revision. Please find our point-to-point response below and highlighted changes in the revised manuscript.

1. The collection efficiency was chosen as 0.5. In fact, the composition dependent CE was more precise. The author should compare these two methods and evaluate whether default CE influence NR-PM$_1$ species quantification.

**Response:**

Thanks for your suggestions. Following the composition-dependent

algorithm brought up by Middlebrook et al. (2012), we evaluated the effect of both aerosol acidity and ammonium nitrate fraction on CE, and further NR-PM$_1$ species quantification.

For acidic aerosols, the CE was calculated as:

$$CE_{dry} = \max\left(0.45, 1.0 - 0.73 \times \left(\frac{NH_4}{NH_{4,pred}}\right)\right), [1]$$

where the NH$_{4,pred}$ was the predicted concentration of ammonium needed to neutralize the inorganic anion mass concentrations observed by the ACSM:

$$NH_{4,pred} = 18 \times \left(\frac{SO_4}{96} \times 2 + \frac{NO_3}{62} + \frac{Chl}{35.45}\right). [2]$$

NH$_4$, SO$_4$, NO$_3$, Chl were the measured aerosol ammonium, sulfate, nitrate, and chloride mass concentrations (in μg m$^{-3}$), respectively. The average ratio of measured NH$_4$ versus predicted NH$_4$ was 0.78, suggesting a weak acidity. The averaged CE in each season (Table AR1) was close to 0.5. However, the year-averaged masses calculated from the acidity dependent CE were 5.7%, 4.9%, 6.3%, 5.3% and 4.8% larger than those from default CE for NO$_3$, SO$_4$, NH$_4$, Chl and organics.

For high ammonium nitrate fraction aerosols, the CE was calculated as:

$$CE_{dry} = \max(0.45, 0.0833 + 0.9167 \times ANMF), [3]$$

where the ANMF was the ammonium nitrate mass fraction:

$$ANMF = \frac{80/62 \times NO_3}{(NH_4 + SO_4 + NO_3 + Chl + Org)}. [4]$$

The Org was the measured organic mass concentrations (in μg m$^{-3}$). The average ANMF was 0.29, which is quite low than the threshold value (0.4) that affects CE (Middlebrook et al., 2012). The ratios of year-averaged masses calculated from the ammonium nitrate mass fraction dependent CE versus those from default CE were 0.96, 1.04, 1.00, 1.01 and 1.03 for NO$_3$,

SO$_4$, NH$_4$, Chl and organics, respectively.

Overall, the biases between NR-PM$_1$ mass concentrations calculated from composition dependent and default CE were acceptable. Thus, we revised the manuscript based on the results presented here.

Table AR1: The aerosol acidity and ammonium nitrate fraction effects on CE. The M$_{CE,acd}$, M$_{CE,anf}$, and M$_{CE,default}$ stand for the average NR-PM$_1$ mass calculated from the acidity dependent CE, the ammonium nitrate mass fraction dependent CE, and default CE.

| | Acidity effect | | | Ammonium nitrate fraction effect | | |
|---|---|---|---|---|---|---|
| | CE | NH$_4$/NH$_{4,pred}$ | M$_{CE,acd}$/M$_{CE,default}$ | CE | ANMF | M$_{CE,anf}$/M$_{CE,default}$ |
| Spring | 0.49 | 0.79 | 1.05 | 0.49 | 0.31 | 1.00 |
| Summer | 0.50 | 0.75 | 1.04 | 0.46 | 0.22 | 1.08 |
| Autumn | 0.48 | 0.83 | 1.08 | 0.47 | 0.26 | 1.06 |
| Winter | 0.49 | 0.77 | 1.05 | 0.51 | 0.38 | 0.93 |
| All | 0.49 | 0.78 | 1.05 | 0.48 | 0.29 | 1.01 |

2. PMF source apportionment was performed for entire study, with two-factor solution being resolved. Considering that the emissions sources could be different in different seasons, PMF should be done separately during each season. Did the author try to do ME-2 analysis with constrained POA profiles to improve results?

**Response:**

Thank you so much for the suggestion. First, we conducted unconstrained PMF source apportionments separately for four seasons. The POA factors were mixed with OOA feature (prominent m/z 44 signal) in 2-factor solutions for all four seasons (Figure AR1-4). Increasing the factor number did not help.

[Figure]

Figure AR1: Mass spectra of 2-4 factor solution from unconstrained PMF for spring.

[Figure]

Figure AR2: Mass spectra of 2-4 factor solution from unconstrained PMF for summer.

[Figure]

Figure AR3: Mass spectra of 2-4 factor solution from unconstrained PMF for autumn.

[Figure]

Figure AR4: Mass spectra of 2-4 factor solution from unconstrained PMF for winter.

Then, we did ME-2 analysis with a priori POA profile from the unconstrained two-factor solution for both four seasons and the entire research period. In ME-2 analysis, a coefficient called a-value was used to

constrain the spectra variation extent of the given priori factor mass spectra (Canonaco et al., 2013). First, the ME-2 analysis (a=0.1) was performed with possible factor number of 3-5 (Figure AR5-8). Besides the POA factor, all the 3-factor solutions split two OOA factors. For most cases, the two OOA factors could be identified as a MO-OOA (more oxidized OOA) and a LO-OOA (less oxidized OOA). Normally, the LO-OOA spectrum is characterized with a relatively higher peak at m/z 43 and a lower O/C ratio or $f_{44}$ than MO-OOA. However, in winter situation, the two OOA factors had no compatible $f_{44/43}$ and $f_{44}$. For example, one OOA factor had higher $f_{44/43}$ but lower $f_{44}$, suggesting failure in clean spiting of OOA factors. As shown in Figure AR5-8, the 4 or 5 factors solutions failed splitting more meaningful factors.

[Figure]

Figure AR5: Mass spectra of 3-5 factor solution from ME-2 analysis (a=0.1) with POA factor constrained for spring. Yellow bar stands for priori POA factor mass spectra from unconstrained 2-factor solution.

[Figure]

Figure AR6: Mass spectra of 3-5 factor solution from ME-2 analysis (a=0.1) with POA factor constrained for summer. Yellow bar stands for priori POA factor mass spectra from unconstrained 2-factor solution.

[Figure]

Figure AR7: Mass spectra of 3-5 factor solution from ME-2 analysis (a=0.1) with POA factor constrained for autumn. Yellow bar stands for priori POA factor mass spectra from unconstrained 2-factor solution.

[Figure]

Figure AR8: Mass spectra of 3-5 factor solution from ME-2 analysis (a=0.1) with POA factor constrained for winter. Yellow bar stands for priori POA factor mass spectra from unconstrained 2-factor solution.

We further conducted 3-factor ME-2 analysis with a value ranging from 0.1 to 0.3 (Figure AR9-10). Similar over split features of OOA factors were found in the other seasons with a value of 0.2 or 0.3. As the ME-2 analysis over the entire period gave consistent results of OOA factor splitting, we compared the time-series of OOA factors with secondary inorganic aerosols. It was interesting to note that the MO-OOA had higher correlations with $NO_3$ (Table AR2), indicating that $NO_3$ came from aged airmasses, possibly regional background sources. Finally, we compared the mass concentrations of ME-2 OA factors with those of unconstrained factors. The square of correlation coefficients ($R^2$) between the two methods were 0.97 and 1.00 for POA and OOA, respectively. A portion of 22.3% of unconstrained POA mass further split into OOA in the ME-2 solution. Overall, the ME-2 results did not change the main conclusions of original manuscript, but possible influences were mentioned accordingly.

[Figure]

Figure AR9: Mass spectra of 3 factor solution from ME-2 analysis (a=0.2) with POA factor constrained for four seasons and the entire study period.

[Figure]

Figure AR10: Mass spectra of 3 factor solution from ME-2 analysis (a=0.3) with POA factor constrained for four seasons and the entire study period.

Table AR2: The correlation coefficients ($R^2$) between OOA factors from ME-2 solution and secondary inorganic aerosols.

| | | a=0.1 | a=0.2 | a=0.3 |
|---|---|---|---|---|
| LO-OOA | $SO_4$ | 0.45 | 0.44 | 0.44 |
| | $NO_3$ | 0.33 | 0.32 | 0.32 |
| MO-OOA | $SO_4$ | 0.30 | 0.31 | 0.32 |
| | $NO_3$ | 0.60 | 0.60 | 0.60 |

3. Simultaneous measurements of chemical compositions at Shanghai tower and the ground benefit the comparisons of vertical differences. In line 140-145, the author compared nitrate at SHT with previous studies. The question is that meteorology significantly impacts PM concentrations. The author simply compared the annual average nitrate contribution with surface measurements, without considering the sampling sites, seasons and meteorology. Another question is why higher nitrate aloft owed to lower temperature. Is it possible that other pathways also contribute to nitrate formation?

**Response:**

Thanks for your suggestion. We summarized $NO_3$ mass fraction related field observations in Shanghai in Table AR3. We found significantly higher $NO_3$ mass fractions in spring and winter at SHT than previous surface studies despite sampling sites, instruments, and years. In summer, the fraction at a rural site (Zhao et al., 2020) was found lower but close to this study. According to Cui et al. (2022), the proportion of $NO_3$ was as high as 26.0% in late autumn of 2018. For a similar period (November) in 2019, the ratio at SHT was 27.2%. We also gathered water-soluble $NO_3$ from MARGA

observations at SUR, further supporting the higher portion of $NO_3$ at SHT. We revised the manuscript accordingly.

The SHT and SRF had impacts from similar emission sources to multi-month time scales. The meteorology seems to be the major factor influencing the higher nitrate at SHT. However, other nitrate formation pathways are also possible. Thus, we revised the relevant sentence and focused on presenting observation results there.

Table AR3: The $NO_3$ mass concentrations and mass fractions in Shanghai.

| Description of sampling sites | Instruments (Particle nature) | Seasons (Year) | $NO_3$ mass concentrations ($\mu g\ m^{-3}$) | $NO_3$ mass fractions (%) | References |
|---|---|---|---|---|---|
| A residential and business area | HR-ToF-AMS (NR-PM$_1$) | Spring (2016) | 3.8±5.7 | 15.9% | Zhu et al. (2021) |
| | | Summer (2016) | 2.9±4.9 | 10.2% | |
| | | Winter (2017) | 7.3±7.0 | 22.9% | |
| A residential and business area | HR-ToF-AMS (NR-PM$_1$) | Spring-early summer (2010) | ~4.8 | 16.3% | Huang et al. (2012) |
| A commercial and residential district | MARGA (Water-soluble PM$_{2.5}$) | Late autumn (2018) | 12.9±12.8 | 27.2% | He et al. (2020) |
| Urban | HR-ToF-AMS (NR-PM$_1$) | Winter (2016) | 7.3±6.9 | 22.1% | Zhu et al. (2018) |
| Urban | HR-ToF-AMS (NR-PM$_1$) | Late autumn (2018) | ~5.6 | 26.0% | Cui et al. (2022) |
| Rural | ACSM (NR-PM$_1$) | Summer (2015) | ~6.5 | 20.0% | Zhao et al. (2020) |
| | | Autumn (2015) | ~9.1 | 22.0% | |

| | | Winter (2015) | ~15.7 | 26.0% | |
| | | Spring (2016) | ~11.2 | 26.0% | |
| A central business district | ACSM (NR-PM$_1$) | Spring (2019) | 4.8±4.8 | 29.9% | This study |
| | | Summer (2019) | 3.3±3.2 | 20.4% | |
| | | Autumn (2019) | 3.4±2.9 | 24.5% | |
| | | Winter (2019) | 7.2±7.6 | 36.8% | |
| A residential and business area | MARGA (Water-soluble PM$_{2.5}$) | Spring (2019) | 7.1 | 24.5% | This study |
| | | Summer (2019) | 4.8 | 19.6% | |
| | | Autumn (2019) | 4.5 | 18.6% | |
| | | Winter (2019) | 12.1 | 27.7% | |

4. The author said throughout the study that SHT is close to the top of PBL but no PBL data was shown. The PBL is dynamically varied during daytime and nighttime, and the PBL height might lower than 600 m under severe haze pollutions.

**Response:**

Thanks for your suggestion. We obtained PBL height (PBLH) at SHT from the nearest ERA5 gridded reanalysis data (Hersbach et al., 2020) (https://cds.climate.copernicus.eu/cdsapp#!/dataset/reanalysis-era5-single-levels?tab=form, accessed 27 November 2022). The ERA-PBLH is calculated utilizing a bulk Richardson method, which was widely used for both convective and stable boundary layers (Kim, 2022). According to Wang

et al. (2018), the ERA data tend to overestimate PBLH at nighttime, but underestimate PBLH during daytime in Eastern China by comparing with PBLH calculated from radiosonde sounding data. Overall, the reanalysis data can capture the diurnal and seasonal cycle of PBL structure.

As shown in Figure AR11, the autumn found the highest PBLH for its prevailing synoptic of the continental high pressure (characterized as weak winds, strong solar radiation, and dry weather), favorable for the PBL development. PBLH in four seasons presented similar diurnal variations. The PBL started to develop at 06:00-08:00 before reaching a daily top at 13:00-14:00, and then decreased until stabilizing after sunset (18:00-19:00). However, the summertime PBL had the longest development period (06:00-19:00), while the wintertime PBL had the shortest (08:00-18:00). At nighttime, the observatory at SHT generally stood on top of stable BL despite the deviations. Whereas the time PBL top reaching SHT site varied during the day. Nevertheless, the PBL had contact with SHT top even for the lower bound of deviation, indicating inevitable mass exchanges between SHT and SUR during the daytime.

Though we agree with you about the possibility of lower PBLH than 600m under severe haze pollutions, the low occurrence of haze events in Shanghai during the study period might not change the pre-mentioned characteristics of PBLH evolution.

[Figure]

Figure AR11: Diurnal variations of the reanalysis PBL height in spring (a), summer (b), autumn (c), and winter (d) at the grid box where the Shanghai Tower (SHT) site is in. The solid line represents the mean value, and the shaded area stands for the standard deviation. The dash lines represent the altitude (~600 m) of the SHT site.

5.  In line 155-165, it is confused that the author described the maximum and minimum temperature and RH in detail solely. In fact, the meteorology was linked to chemical compositions. Discussing the influence and interaction between meteorology and PM is more charming. Why the author showed the daily average values in Table 1?

**Response:**

We totally agree that the meteorology is closely linked to chemical compositions. We described the maximum and minimum temperature and RH to give a basic concept of the differences between SHT and SUR meteorological environments. More discussion can be found in section 3.3.2. We modified the manuscript to clarify.

We also provided hourly average values in Table AR4, where the statistics had virtually identical mean values to those in Table 1, while larger standard deviations, especially for aerosol species. As we discussed seasonal features of aerosol and meteorology in section 3.1, the deviations of daily average values excluded the effects of intraday fluctuations.

Table AR4: The seasonal and annual averaged concentrations of aerosol species ($\mu g\ m^{-3}$) and meteorological parameters.

| | | Spring | Summer | Autumn | Winter | Annual |
|---|---|---|---|---|---|---|
| **Aerosol Species ($\mu g\ m^{-3}$)** | | | | | | |
| SHT | $PM_1$ | 18.5±13.9 | 16.9±14.3 | 14.7±11.4 | 19.7±17.6 | 17.4±14.6 |
| | $PM_{2.5}$ | 25.4±17.9 | 22.8±17.6 | 22.1±17.0 | 31.9±28.8 | 25.7±21.6 |
| | $SO_4$ | 3.0±2.4 | 4.3±2.7 | 3.1±2.1 | 3.3±2.7 | 3.4±2.6 |
| | $NO_3$ | 4.8±6.3 | 3.4±4.3 | 3.4±4.0 | 7.2±8.7 | 4.7±6.4 |
| | $NH_4$ | 2.0±1.9 | 2.0±1.6 | 1.9±1.4 | 2.6±2.6 | 2.1±2.0 |
| | Chl | 0.2±0.3 | 0.2±0.2 | 0.3±0.2 | 0.4±0.5 | 0.3±0.3 |
| | OA | 6.0±4.9 | 6.6±6.3 | 5.0±3.8 | 5.8±4.9 | 5.9±5.2 |
| | POA | 1.9±1.9 | 2.4±2.6 | 1.4±1.4 | 1.7±1.6 | 1.9±2.0 |
| | OOA | 4.1±3.2 | 4.3±3.8 | 3.6±2.6 | 4.2±3.5 | 4.0±3.4 |
| SUR | $PM_{2.5}$ | 29.1±19.7 | 24.7±16.0 | 24.2±18.1 | 43.6±33.2 | 30.4±24.1 |
| **Meteorological parameters** | | | | | | |
| SHT | T (°C) | 13.3±5.6 | 22.8±3.5 | 15.9±4.9 | 5.9±4.1 | 14.5±7.6 |
| | RH (%) | 61.1±23.6 | 79.6±11.4 | 74.9±13.8 | 72.1±17.3 | 71.9±18.5 |
| SUR | T (°C) | 16.2±5.6 | 26.5±3.9 | 19.7±5.5 | 8.6±4.1 | 17.7±8.1 |
| | RH (%) | 71.0±23.6 | 82.8±14.8 | 76.7±18.6 | 77.5±20.4 | 77.0±20.1 |

6. In Sec. 3.2.2, please give the definition of anomaly. Is it calculated by comparing with annual average or history records? Why were they anomaly?

**Response:**

The anomaly was the monthly deviation from annual average. By calculating the anomaly, we intended to find monthly changes relative to the whole year. The comparison of SHT and SUR PM anomalies allows us to see the consistency of monthly features at two altitudes.

The manuscript was modified to clarify.

7. In line 202-203, the author state that extra aerosol productions contributed to higher $PM_{2.5}$ concentrations at SHT than surface. Please elaborate the conclusion.

**Response:**

Given that SHT was farther from the direct emission sources than SUR, the $PM_{2.5}$ at SHT tended to have lower concentration than SUR as in the other months despite vertical mixing during the daytime. Thus, the higher $PM_{2.5}$ at SHT in August indicated extra aerosol productions at mid-upper PBL.

8. In line 205-210, the author said that exchange between SHT and surface only exits in daytime. If nocturnal PBL is higher than SHT, nighttime exchange can also occur. Nighttime $PM_{2.5}$ at SHT was not independent from the ground level.

**Response:**

As discussed in the response of question 4, it's true that the exchange between SHT and surface only exits in daytime, at least in the view of ERA5-PBLH. However, we acknowledge that the PBLH is crucial for the vertical structure analysis, and direct observations of PBLH are in need to give precise view of concurrent boundary layer processes. We mentioned this factor in the conclusion part.

9. In Sec. 3.2.4 and figure 4, the author can also plot and discuss mass fractions of $NR-PM_1$ during daytime and nighttime separately.

**Response:**

Thanks for your suggestion, and figure 4 was revised and discussed accordingly:

The daytime and nighttime mass fractions were shown in Figure AR12. As results of vertical mixing, the larger portions of primary species (POA and Chl) during daytime were notable, especially for summer and autumn. The changes of OOA, NO$_3$, and NH$_4$ were slight, with increase of OOA and NH$_4$, but decrease of NO$_3$ from nighttime to daytime. Accordingly, SO$_4$ saw lower fraction in NR-PM$_1$ during the daytime. More diurnal features of NR-PM$_1$ can be found in section 3.3.4.

But as you mentioned in question 3, "simultaneous measurements of chemical compositions at Shanghai tower and the ground benefit the comparisons of vertical differences". The lack of surface measurements of chemical compositions prevented us from digging more into the vertical differences between SHT and SUR.

[Figure]

Figure AR12: The monthly averaged (a) and seasonal averaged (b-d) mass fractions (%) of NR-PM$_1$ at SHT. The mass fractions (%) are calculated based on all (b), daytime (c), and nighttime (d) data. The monthly averaged mass concentrations ($\mu$g m$^{-3}$) of NR-PM$_1$ are also shown. The solid and dashed line represent SHARP PM$_1$ and NR-PM$_1$, respectively.

10. In figure 6, NO$_2$ at SHT increased by 21.8-61.4% from 08:00 to 12:00, while they were reduced at ground level during this time. Thus, vertical mixing could be the explanation rather than vehicles. In fact, the peak at morning at surface was attributed to traffic during morning rush hour.

**Response:**

Thank you for the note. The corresponding sentences were corrected.

11. In line 62, "vatical distribution" might be a typo.

**Response:**

Thank you. It was revised to 'vertical distribution'.

12. Please uniform the subscripts of sulfate, nitrate and ammonium throughout the study.

**Response:**

Thank you. We rechecked the relevant subscripts.

**References:**

Canonaco, F., Crippa, M., Slowik, J. G., Baltensperger, U., and Prevot, A. S. H.: SoFi, an IGOR-based interface for the efficient use of the generalized multilinear engine (ME-2) for the source

apportionment: ME-2 application to aerosol mass spectrometer data, Atmospheric Measurement Techniques, 6, 3649-3661, 10.5194/amt-6-3649-2013, 2013.

Cui, S. J., Huang, D. D., Wu, Y. Z., Wang, J. F., Shen, F. Z., Xian, J. K., Zhang, Y. J., Wang, H. L., Huang, C., Liao, H., and Ge, X. L.: Chemical properties, sources and size-resolved hygroscopicity of submicron black-carbon-containing aerosols in urban Shanghai, Atmospheric Chemistry and Physics, 22, 8073-8096, 10.5194/acp-22-8073-2022, 2022.

He, X., Wang, Q. Q., Huang, X. H. H., Huang, D. D., Zhou, M., Qiao, L. P., Zhu, S. H., Ma, Y. G., Wang, H. L., Li, L., Huang, C., Xu, W., Worsnop, D. R., Goldstein, A. H., and Yu, J. Z.: Hourly measurements of organic molecular markers in urban Shanghai, China: Observation of enhanced formation of secondary organic aerosol during particulate matter episodic periods, Atmospheric Environment, 240, 10.1016/j.atmosenv.2020.117807, 2020.

Hersbach, H., Bell, B., Berrisford, P., Hirahara, S., Horanyi, A., Munoz-Sabater, J., Nicolas, J., Peubey, C., Radu, R., Schepers, D., Simmons, A., Soci, C., Abdalla, S., Abellan, X., Balsamo, G., Bechtold, P., Biavati, G., Bidlot, J., Bonavita, M., De Chiara, G., Dahlgren, P., Dee, D., Diamantakis, M., Dragani, R., Flemming, J., Forbes, R., Fuentes, M., Geer, A., Haimberger, L., Healy, S., Hogan, R. J., Holm, E., Janiskova, M., Keeley, S., Laloyaux, P., Lopez, P., Lupu, C., Radnoti, G., de Rosnay, P., Rozum, I., Vamborg, F., Villaume, S., and Thepaut, J. N.: The ERA5 global reanalysis, Quarterly Journal of the Royal Meteorological Society, 146, 1999-2049, 10.1002/qj.3803, 2020.

Huang, X. F., He, L. Y., Xue, L., Sun, T. L., Zeng, L. W., Gong, Z. H., Hu, M., and Zhu, T.: Highly time-resolved chemical characterization of atmospheric fine particles during 2010 Shanghai World Expo, Atmospheric Chemistry and Physics, 12, 4897-4907, 10.5194/acp-12-4897-2012, 2012.

Kim, K.-Y.: Diurnal and seasonal variation of planetary boundary layer height over East Asia and its climatic change as seen in the ERA-5 reanalysis data, SN Applied Sciences, 4, 39, 10.1007/s42452-021-04918-5, 2022.

Middlebrook, A. M., Bahreini, R., Jimenez, J. L., and Canagaratna, M. R.: Evaluation of composition-dependent collection efficiencies for the Aerodyne aerosol mass spectrometer using field data, Aerosol Sci. Technol., 46, 258-271, 10.1080/02786826.2011.620041, 2012.

Wang, Y. J., Xu, X. D., Zhao, Y., and Wang, M. Z.: Variation characteristics of the planetary boundary layer height and its relationship with PM2.5 concentration over China, Journal of Tropical Meteorology, 24, 385-394, 10.16555/j.1006-8775.2018.03.011, 2018.

Zhao, Q. B., Huo, J. T., Yang, X., Fu, Q. Y., Duan, Y. S., Liu, Y. X., Lin, Y. F., and Zhang, Q.: Chemical characterization and source identification of submicron aerosols from a year-long real-time observation at a rural site of Shanghai using an Aerosol Chemical Speciation Monitor, Atmospheric Research, 246, 10.1016/j.atmosres.2020.105154, 2020.

Zhu, W., Zhou, M., Cheng, Z., Yan, N., Huang, C., Qiao, L., Wang, H., Liu, Y., Lou, S., and Guo, S.: Seasonal variation of aerosol compositions in Shanghai, China: Insights from particle aerosol mass spectrometer observations, Science of The Total Environment, 771, 144948, 10.1016/j.scitotenv.2021.144948, 2021.

Zhu, W. F., Xie, J. K., Cheng, Z., Lou, S. R., Luo, L. N., Hu, W. W., Zheng, J., Yan, N. Q., and Brooks, B.: Influence of chemical size distribution on optical properties for ambient submicron particles during severe haze events, Atmospheric Environment, 191, 162-171, 10.1016/j.atmosenv.2018.08.003, 2018.

---

## Author Comment (AC2)

Dear Editor,

We appreciate the prompt reviews and would like to thank the two reviewers for insightful comments and suggestions on our manuscript entitled "Characteristics of fine particle matters at the top of Shanghai Tower" (MS No.: egusphere-2022-782). We have carefully considered all comments and suggestions. Listed below are our point-by-point responses to all comments and suggestions of this reviewer (Reviewer's points in black, our responses in blue).

**Anonymous Referee #2**

In this manuscript, the authors gave a very details analysis of observed one year continuous $PM_{2.5}$ and its chemical components at the top of 632 m high Shanghai Tower (SHT). The data collected were precious, and the topic is of great interesting to recognize vertical $PM_{2.5}$ characteristics and its formation processes related to emission, chemical production and boundary layer (BL) etc. The analysis is mostly sound, but some details need clarify. I recommend a minor revision and my specific comments listed below.

**Response:**

We sincerely thank the reviewer for the valuable comments. These comments have been carefully addressed during revision. Please find our point-to-point response below and highlighted changes in the revised manuscript.

Specific comments:

1.  My primary concern is that the study address the $PM_{2.5}$ and its chemical components at SHT dominating by vertical mixing from surface (most in

daytime) and chemical production therein from surface precursors, while omitted the PM originating from transport outside Shanghai. The seasonal winds induced by Asia monsoon are quite difference in upstream (ocean or land, most natural or anthropogenic in background) and could impact much at SHT than on the surface. I suggest the authors should refer to this factor or indicting for future research.

**Response:**

Thanks for your constructive suggestion. We analyzed the transport pathway at the height of 100 m and 600 m in each season, using 72 h back trajectory from HYbrid Single Particle Lagrangian Integrated Trajectory (HYSPLIT) model. Though the two heights had similar tracks (Figure AR1), the small departures might lead to different source origins. Thus, we further calculated the $R^2$ between $PM_{2.5}$ at SHT and SUR based on hourly and daily-averaged data (Figure AR2). We found that the $R^2$ increased by 0.12, 0.29, 0.20, and 0.13 on average for spring, summer, autumn, and winter from hourly data to daily-averaged data. The pronounced increase for $R^2$ indicates that the differences of $PM_{2.5}$ between SUR and 600m mostly came from the subdaily variations.

As suggested, we refer to the influences of regional transport in the conclusion part.

[Figure]

Figure AR1: Air transport pathway at the height of 100 m (solid lines) and 600 m (dashed lines) during spring (a), summer (b), autumn (c), and winter (d). The 72h back trajectory was simulated using HYbrid Single Particle Lagrangian Integrated Trajectory (HYSPLIT) model with the location of Shanghai Tower as central coordinate.

[Figure]

Figure AR2: Monthly variations of correlation coefficients ($R^2$) between PM$_{2.5}$ at SHT and SUR. The dashed line represents $R^2$ based on hourly PM$_{2.5}$, and the solid line for daily-averaged PM$_{2.5}$.

2. nitrate ($NO_3^-$) and sulfate ($SO_4^{2-}$) should be correctly present in the manuscript.

**Response:**

Thank you for your suggestion. Though the measurements of nitrate and sulfate are relying on ionized fragments from Q-ACSM, both abbreviates of $NO_3$ ($SO_4$) and $NO_3^-$ ($SO_4^{2-}$) were wildly used in previous studies (e.g., Cao et al., 2019; Zhang et al., 2011; Ng et al., 2011; Zhou et al., 2018). We uniformed the subscripts of nitrate and sulfate as $NO_3$ and $SO_4$ throughout the study.

3. In line 69, Shanghai is not only one of the most densely populated megacities in China, but also in the world.

**Response:**

Thanks for your suggestion. Modifications were made accordingly.

4. In this study, the heights of BL were important. Please give a brief introduction of BLHs in different season and day and night in Shanghai.

**Response:**

Thanks for your suggestion. We obtained PBL height (PBLH) at SHT from the nearest ERA5 gridded reanalysis data (Hersbach et al., 2020) (https://cds.climate.copernicus.eu/cdsapp#!/dataset/reanalysis-era5-singlelevels?tab=form, accessed 27 November 2022). The ERA-PBLH is calculated utilizing a bulk Richardson method, which was widely used for both convective and stable boundary layers (Kim, 2022). According to Wang et al. (2018), the ERA data tend to overestimate PBLH at nighttime, but underestimate PBLH during daytime in Eastern China by comparing with PBLH calculated from radiosonde sounding data. Overall, the reanalysis data can capture the diurnal and seasonal cycle of PBL structure.

As shown in Figure AR3, the autumn found the highest PBLH for its prevailing synoptic of the continental high pressure (characterized as weak winds, strong solar radiation, and dry weather), favorable for the PBL development. PBLH in four seasons presented similar diurnal variations. The PBL started to develop at 06:00-08:00 before reaching a daily top at 13:00-14:00, and then decreased until stabilizing after sunset (18:00-19:00). However, the summertime PBL had the longest development period (06:00-19:00), while the wintertime PBL had the shortest (08:00-18:00). At nighttime, the observatory at SHT generally stood on top of stable BL (SBL) despite the deviations. Whereas the time PBL top reaching SHT site varied during the day. Nevertheless, the PBL had contact with SHT top even for the lower bound of deviation, indicating inevitable mass exchanges between SHT and SUR during the daytime.

Modifications were made accordingly.

[Figure]

Figure AR3: Diurnal variations of the reanalysis PBL height in spring (a), summer (b), autumn (c), and winter (d) at the grid box where the Shanghai Tower (SHT) site is in. The solid line represents the mean value, and the shaded area stands for the standard deviation. The dash lines represent the altitude (~600 m) of the SHT site.

5. In line 176-178, the inference is not very exact. The seasonal variations of BLH could be key factor for the similar monthly variations of $PM_{2.5}$ at SHT and SUR, and related to regional transport, vertical diffusions etc. And I am happy to find you mentioned of regional transport, while did not raise this in conclusion, abstract and other paragraphs.

**Response:**

We totally agree that the seasonal variation of BLH was a key factor for the monthly variations of $PM_{2.5}$ at SUR. However, the shallower (deeper) BLH would lead to less (more) mass exchanges between SUR and SHT, resulting in lower (higher) mass concentrations transported from surface to high altitude. Thus, we concluded that the similar monthly variations of $PM_{2.5}$ at

SHT and SUR were more likely related to regional transport and local emissions.

6. In 188-190, the anomalies may reflect the seasonal variations of BLHs.

**Response:**

Thanks again for your suggestions regarding seasonal behaviors of PM. As mentioned in question 5, the seasonal variations of BLHs had opposite impacts on SHT and SUR. However, the shallower BLH meant less contact time between SHT and SUR air in winter, presenting larger differences between the anomalies at two altitudes.

7. In 210-211, "completely" is not very exact because in some synoptic conditions, the mass and energy exchange between free troposphere and within the BL could occur.

**Response:**

Thanks for noting. We changed "completely" to "mostly".

8. What were the definitions of POA, OOA, and HOA, and their chemical components in this study?

**Response:**

The POA and OOA were two factors we retrieved from PMF analysis. As the profile of POA was a mixture of HOA, COA, and CCOA features, which were identified as primary components (Duan et al., 2019). the HOA profile is recognized by noticeable hydrocarbon ion series of $C_nH_{2n-1}$ and $C_nH_{2n+1}$; particularly m/z 27, 29, 41, 43, 55, 57, 67 and 71. COA is characterized by

higher ratio of m/z 55 than m/z 57, and CCOA mass spectrum is acknowledged as distinctive polycyclic aromatic hydrocarbons (PAHs) fragments. OOA profile sees prominent ion fragment at m/z 44 ($CO_2^+$). Modifications were made accordingly.

9. In figure 5, why there was the largest difference of $PM_{2.5}$ between SHT&SUR in summer, while the largest difference of NOR in winter and spring in figure 7?

**Response:**

Thanks for your suggestion. We gathered Monitor for AeRosols and Gases in ambient (MARGA) data from Pudong New District Environmental Monitoring Station to calculate the NOR at surface. As the largest differences of $PM_{2.5}$ between SHT and SUR were found around noon, the NOR during the daytime were calculated accordingly. As shown in Table AR1, the NOR was higher at SHT than SUR in spring, summer, and autumn, while lower at SHT in winter. Besides, the most significant difference of NOR appeared in summer, when the largest $PM_{2.5}$ departures between two altitudes were found.

Table AR1: The NOR during the daytime at SHT and SUR in four seasons. The NOR was calculated as: $[NO_3]/([NO_3]+[NO_2])$, where [x] points to the molar concentration of x.

|      | Spring | Summer | Autumn | Winter |
|------|--------|--------|--------|--------|
| SHT  | 0.13   | 0.10   | 0.09   | 0.15   |
| SUR  | 0.12   | 0.07   | 0.07   | 0.16   |

10. In line 370, latitude should be altitude.

**Response:**

Thank you for the note. Revised.

11. In line 374, "since the SO₂ level was relatively lower than the other seasons.". or also because the favorable diffusion and wet scavenging condition of atmosphere in summer.

**Response:**

Thank you. It was revised as suggested.

**References:**

Cao, L. M., Huang, X. F., Wang, C., Zhu, Q., and He, L. Y.: Characterization of submicron aerosol volatility in the regional atmosphere in Southern China, Chemosphere, 236, 11, 10.1016/j.chemosphere.2019.124383, 2019.

Duan, J., Huang, R. J., Lin, C. S., Dai, W. T., Wang, M., Gu, Y. F., Wang, Y., Zhong, H. B., Zheng, Y., Ni, H. Y., Dusek, U., Chen, Y., Li, Y. J., Chen, Q., Worsnop, D. R., O'Dowd, C. D., and Cao, J. J.: Distinctions in source regions and formation mechanisms of secondary aerosol in Beijing from summer to winter, Atmospheric Chemistry and Physics, 19, 10319-10334, 10.5194/acp-19-10319-2019, 2019.

Hersbach, H., Bell, B., Berrisford, P., Hirahara, S., Horanyi, A., Munoz-Sabater, J., Nicolas, J., Peubey, C., Radu, R., Schepers, D., Simmons, A., Soci, C., Abdalla, S., Abellan, X., Balsamo, G., Bechtold, P., Biavati, G., Bidlot, J., Bonavita, M., De Chiara, G., Dahlgren, P., Dee, D., Diamantakis, M., Dragani, R., Flemming, J., Forbes, R., Fuentes, M., Geer, A., Haimberger, L., Healy, S., Hogan, R. J., Holm, E., Janiskova, M., Keeley, S., Laloyaux, P., Lopez, P., Lupu, C., Radnoti, G., de Rosnay, P., Rozum, I., Vamborg, F., Villaume, S., and Thepaut, J. N.: The ERA5 global reanalysis, Quarterly Journal of the Royal Meteorological Society, 146, 1999-2049, 10.1002/qj.3803, 2020.

Kim, K.-Y.: Diurnal and seasonal variation of planetary boundary layer height over East Asia and its climatic change as seen in the ERA-5 reanalysis data, SN Applied Sciences, 4, 39, 10.1007/s42452-021-04918-5, 2022.

Ng, N. L., Herndon, S. C., Trimborn, A., Canagaratna, M. R., Croteau, P. L., Onasch, T. B., Sueper, D., Worsnop, D. R., Zhang, Q., Sun, Y. L., and Jayne, J. T.: An aerosol chemical speciation monitor (ACSM) for routine monitoring of the composition and mass concentrations of ambient aerosol, Aerosol Sci. Technol., 45, 780-794, 10.1080/02786826.2011.560211, 2011.

Wang, Y. J., Xu, X. D., Zhao, Y., and Wang, M. Z.: Variation characteristics of the planetary boundary layer height and its relationship with PM2.5 concentration over China, Journal of Tropical

Meteorology, 24, 385-394, 10.16555/j.1006-8775.2018.03.011, 2018.

Zhang, Q., Jimenez, J. L., Canagaratna, M. R., Ulbrich, I. M., Ng, N. L., Worsnop, D. R., and Sun, Y. L.: Understanding atmospheric organic aerosols via factor analysis of aerosol mass spectrometry: a review, Anal. Bioanal. Chem., 401, 3045-3067, 10.1007/s00216-011-5355-y, 2011.

Zhou, W., Sun, Y. L., Xu, W. Q., Zhao, X. J., Wang, Q. Q., Tang, G. Q., Zhou, L. B., Chen, C., Du, W., Zhao, J., Xie, C. H., Fu, P. Q., and Wang, Z. F.: Vertical Characterization of Aerosol Particle Composition in Beijing, China: Insights From 3-Month Measurements With Two Aerosol Mass Spectrometers, Journal of Geophysical Research-Atmospheres, 123, 13016-13029, 10.1029/2018jd029337, 2018.

---

## Author Response (AR2)

Dear Editor,

We appreciate the prompt reviews and thanks a lot for your constructive suggestions on our manuscript entitled "Characteristics of fine particle matters at the top of Shanghai Tower" (MS No.: egusphere-2022-782). We have carefully considered all comments and suggestions. Listed below are our point-by-point responses (Your points in black, our responses in blue).

Thank you for your thorough and comprehensive response to the referee comments. I find that the manuscript is significantly approved and am pleased to accept it for publication following attention to the minor comments outlined below. Line numbers refer to the track changes version of the manuscript unless otherwise noted.

**Response:**

Thank you again for your valuable comments, which have been carefully addressed during revision. Please find our point-to-point response below and highlighted changes in the revised manuscript. Line numbers refer to the track changes version of the manuscript.

1. Lines 109-112: Please add a note saying that a composition dependent collection efficiency was investigated and resulted in no significant changes – something along the response that was included to the referee's question.

**Response:**

Thanks for your suggestion. Add accordingly. Please see lines 112-113.

2. Line 182: I am confused about where and when the MARGA

measurements were made. Please elaborate on these.

**Response:**

The MARGA measurements were operational at PEMC site, but not available for public access. The MARGA and SHT $NO_3$ data were compared for the exact same period. Please see lines 192-193.

3. Section 3.23: I think the organization and presentation of information in this section should be modified to improve readability. In particular, after reading the first paragraph in the section, I was confused by the statement that "the lowest RPC in winter could be attributed to the shallowest PBL height." It is lowest because all the RPC values are negative and thus even though it is the lowest, it represents the largest change between aloft and surface measurements. If both measurements are within the PBL, I would expect a shallower PBL to lead to a shallower gradient. It is clarified in the second paragraph though that the result is really driven by the differences between day and night. I think harmonizing these two discussions to make this clear would benefit the reader.

**Response:**

Thank you for your suggestion. In the first paragraph of Section 3.2.3, we speculate that lower $PM_{2.5}$ observed at SHT than SUR in winter would be partially attributed to the relatively weak vertical diffusion of $PM_{2.5}$ from surface to high latitude in addition to the differences between day and night. However, we agree your comments to avoid possible confusion. Therefore, we just present the RPC results in the first paragraph. The confusing statement was removed. Please lines 253-254, and 271.

4. Lines 370-372 and Figure 8: I don't agree with the statement that the

"diurnal cycle of NOR kept roughly stable" since the magnitude the max-min change for NOR, particularly in winter and spring, is ~0.04 while for SOR it is ~0.06. These numbers are not that different. I think it is more the fact that the variation in NOR is not reproducible between the seasons and is not as straightforward to interpret the variation as the SOR is. I think adding some clarifying text about the subtly of this point would be helpful to the reader. I also suggest adding shading indicating the standard deviations as well as a comment in the figure caption indicating what quantity is plotted (mean, median, etc.). It could be once variability is included that there really isn't variation in NOR, but with the information presented, the reader is not able to judge that.

**Response:**

Thanks for your suggestion. According to the definition of NOR ($[NO_3]/([NO_3]+[NO_2])$), the increase of $NO_2$ would lead to the decrease of NOR. However, the NOR at SHT did not see notable decrease from 8:00 to 12:00, when $NO_2$ increased significantly by 21.8-61.4%. Apparently, the statement "diurnal cycle of NOR kept roughly stable" brought confusion. Therefore, the statement and Figure 8 (Figure AR1) were revised as suggested. Please see lines 354, 377-386.

[Figure]

Figure AR1: Diurnal variations of SOR and NOR observed at SHT (blue line) in four seasons and the entire observation period. The line stands for the mean values. The shaded area represents the standard deviation.

5. Lines 471-476: I think this discussion should be moved much earlier in the manuscript as it is an important point for the reader to keep in mind when thinking about the results. I recommend including it at the beginning of Section 3 (before Sect. 3.1) as an overall comment on the interpretation used throughout the section. I think it would also be useful to include the discussion and the figures regarding the back trajectory analysis that were provided in the response to referee document. Placing these added figures and discussion in the supporting information is appropriate.

**Response:**

Thanks a lot for your suggestion. The discussion was moved, and the back trajectory analysis was included. Please see lines 159-165, and 482-486.

Technical:

1. Please consider placing Fig. S1 in the main text so that the reader can follow along more easily with the discussion surrounding the HOA and OOA attribution.

**Response:**

Thanks for noting. Revised.

2. "$m/z$" should be italicized throughout the text

**Response:**

Thanks for pointing out. Revised accordingly.

3. Line 178: Table S1 is missing from the supporting information. Please add it to the document.

**Response:**

Thanks for noting. Table S1 was added in supporting information.

4. Line 364: vegetables --> vegetation

**Response:**

Thanks for noting. Revised.

---

## Author Response (AR3)

**Reply to comments on "Characteristics of fine particle matters at the top of Shanghai Tower"** *by Changqin Yin, Jianming Xu, Wei Gao, et al.*

Dear editor, thank you very much for the supportive feedback and consideration of our paper for final publication in ACP, subject to technical corrections. Our point-to-point replies to the technical corrections are listed below (Your points in black, our responses in blue):

**Comments to the author:**

Congratulations! I am pleased to accept the manuscript subject to the technical corrections identified by the editorial office.

**Notification to the authors:**

1. Regarding your figure 1 (photos): with the next revision, please check if a copyright statement/image credit is required and add it to the figure caption, if applicable. If you are the originator of the photos, you can just inform us via email.

**Response:**

Thanks for noting. We are the originator of the photos in figure 1. In addition, we added a copyright statement for the map images in the figure caption:

"Figure 1: The deployment of SHT site. The upper-left image gives a view of Yangtze River Delta Region (screenshot from Google Earth 2022 map data: Data SIO, NOAA, U.S. Navy, NGA, GEBCO. Image Landsat/Copernicus). The upper-right image shows the map of sample sites (Image © 2022 Maxar Technologies). The red star in the lower-left photo denotes the platform at the top of SHT."

2. Please ensure that the colour schemes used in your maps and charts allow readers with colour vision deficiencies to correctly interpret your findings. Please check your figures using the Coblis – Color Blindness Simulator (https://www.color-blindness.com/coblis-color-blindness-simulator/) and revise the colour schemes accordingly.

**Response:**

Thanks for noting. We checked the figures in the manuscript. Figures 1-5, 7-10 are fine. We changed the color of $NH_4$ from dark orange to a lighter one for both Figure 6 and Figure 11. The legends for the lines in Figure 6a was removed to avoid confusion for monochromacy/achromatopsia, as the lines are easy to distinguish from the figure caption. Besides, the color of $NH_4$ for Figure S10 in the supplementary materials was changed accordingly.